# DiCache: Let Diffusion Model Determine Its Own Cache

**Jiazi Bu**[1,5*]  **Pengyang Ling**[2,5*]  **Yujie Zhou**[1,5*]  **Yibin Wang**[3,6]
**Yuhang Zang**[5]  **Dahua Lin**[4,5,7]  **Jiaqi Wang**[5,6†]
[1]Shanghai Jiao Tong University  [2]University of Science and Technology of China
[3]Fudan University  [4]The Chinese University of Hong Kong  [5]Shanghai AI Laboratory
[6]Shanghai Innovation Institute  [7]CPII under InnoHK
`https://bujiazi.github.io/dicache.github.io/`

## Abstract

Recent years have witnessed the rapid development of acceleration techniques for diffusion models, especially caching-based acceleration methods. These studies seek to answer two fundamental questions: *"When to cache"* and *"How to use cache"*, typically relying on predefined empirical laws or dataset-level priors to determine caching timings and adopting handcrafted rules for multi-step cache utilization. However, given the highly dynamic nature of the diffusion process, they often exhibit limited generalizability and fail to cope with diverse samples. In this paper, a strong sample-specific correlation is revealed between the variation patterns of the shallow-layer feature differences in the diffusion model and those of deep-layer features. Moreover, we have observed that the features from different model layers form similar trajectories. Based on these observations, we present **DiCache**, a novel training-free adaptive caching strategy for accelerating diffusion models at runtime, answering both when and how to cache within a unified framework. Specifically, DiCache is composed of two principal components: (1) *Online Probe Profiling Scheme* leverages a shallow-layer online probe to obtain an on-the-fly indicator for the caching error in real time, enabling the model to dynamically customize the caching schedule for each sample. (2) *Dynamic Cache Trajectory Alignment* adaptively approximates the deep-layer feature output from multi-step historical caches based on the shallow-layer feature trajectory, facilitating higher visual quality. Extensive experiments validate DiCache's capability in achieving higher efficiency and improved fidelity over state-of-the-art approaches on various leading diffusion models including WAN 2.1, HunyuanVideo and Flux. Our code is available at `https://github.com/Bujiazi/DiCache`.

## 1 Introduction

Over the past few years, diffusion models (Song & Ermon, 2019; Sohl-Dickstein et al., 2015; Ho et al., 2020; Dhariwal & Nichol, 2021) have markedly advanced the frontiers of visual synthesis. Initially grounded in the lightweight U-Net (Ronneberger et al., 2015) architecture, diffusion models have achieved substantial progress in both image (Rombach et al., 2022; Podell et al., 2023) and video synthesis Blattmann et al. (2023); Guo et al. (2023); Chen et al. (2024a). Recent efforts (Esser et al., 2024; Labs, 2024; Chen et al., 2023) have integrated transformer-based architectures into diffusion models for larger model capacity and improved performance, especially in the field of video generation (Kong et al., 2024; Wan et al., 2025; HaCohen et al., 2024; Yang et al., 2024). Despite their effectiveness, diffusion models often suffer from substantial inference costs and low generation speed, primarily stemming from the rapid expansion in model scale and intricacy.

To mitigate these limitations, numerous approaches (Lu et al., 2022; Li et al., 2024; Liu et al., 2025a) on accelerating diffusion models have been proposed. Most training-based acceleration strategies (Meng et al., 2023; Sauer et al., 2024; Wang et al., 2023; Chen et al., 2025; Ma et al.,

---

*Equal contribution. †Corresponding author.

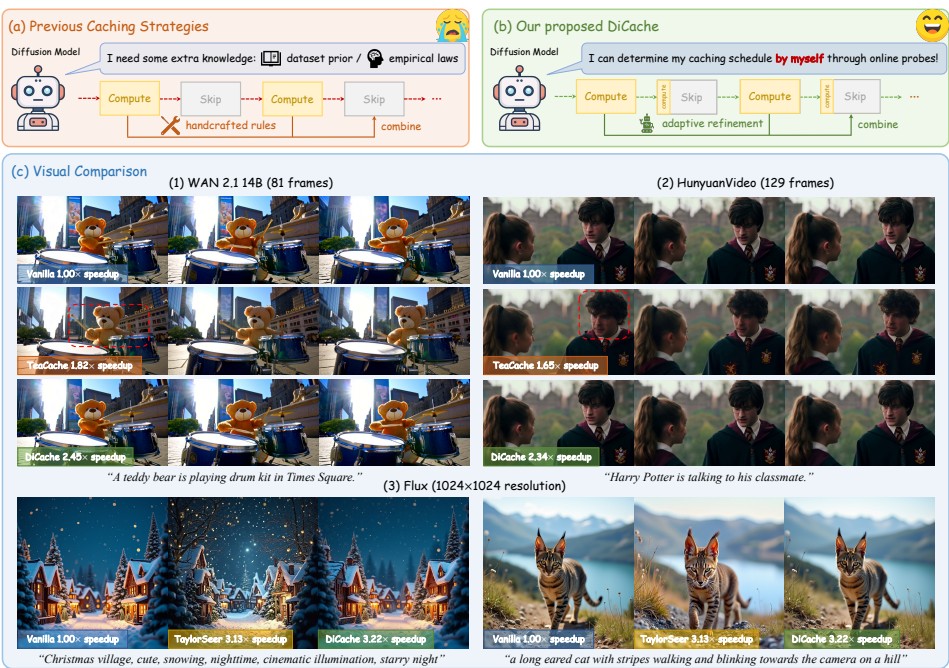

Figure 1: **Comparison between our proposed DiCache and previous caching methods.** (a) Previous studies typically rely on dataset priors or empirical laws to skip timesteps, and resort to handcrafted rules to utilize multi-step caches. (b) DiCache employs an online probe to adaptively adjust its caching strategy at runtime. (c) A brief visual comparison between DiCache and existing state-of-the-art methods, in which DiCache demonstrates superiority in both quality and efficiency.

2024; Li et al., 2024) entail significant training costs and require supplementary training data, which are less suitable for broad deployment. Therefore, recent studies have focused on training-free acceleration methods (Zhang et al., 2025b; Xi et al., 2025; Ren et al., 2025; Lu et al., 2022; Liu et al., 2025a). Among them, caching-based methods (Chen et al., 2024b; Selvaraju et al., 2024; Xu et al., 2018; Zhao et al., 2024; Lv et al., 2024) provide a lightweight choice for diffusion model acceleration by leveraging the similarity between features at consecutive timesteps and reusing them strategically to reduce computation redundancy. Early uniform caching strategies reuse cached features at fixed intervals, which fail to adapt to the diffusion process's time-varying behaviors, resulting in low inference speed and degraded visual quality. Recently, AdaCache (Kahatapitiya et al., 2024) suggests dynamically modifying the cache interval based on content complexity. TeaCache (Liu et al., 2025a) employs a polynomial function calibrated on offline datasets to determine its cache schedule. EasyCache (Zhou et al., 2025) replaces this polynomial estimator with an empirical transformation rate law. TaylorSeer (Liu et al., 2025b) leverages multi-step caches under a Taylor expansion-based forecast mechanism to predict current features for enhanced visual texture and details.

Caching-based acceleration confronts two core issues: *"When to cache"* and *"How to use cache"*. Existing methods address these questions by (1) relying on predefined empirical laws or dataset-level priors to make their cache schedules, and (2) resorting to manually crafted rules for multi-step cache utilization, as shown in Fig. 1 (a). These two key limitations hinder their flexibility to adapt to diverse samples, leading to suboptimal inference efficiency and reduced similarity regarding original results, as illustrated in Fig. 1 (c). Therefore, an adaptive paradigm wherein *the model autonomously determines its reuse strategy without recourse to external empirical priors* is imperative.

In this paper, we uncover that (1) **for a given sampling process, the difference in shallow-layer features strongly correlates with that in deep-layer features on a sample-specific basis** (Fig. 3), enabling them to serve as an on-the-fly proxy for the final model output evolution. Since the optimal moment to reuse cached features is governed by the difference between model outputs at consecutive timesteps (Liu et al., 2025a), it is possible to employ an online shallow-layer probe to efficiently obtain an indicator of output changes at runtime, thereby adaptively adjusting the caching strategy for each individual sample. (2) **the features from different DiT blocks form similar trajectories**

(Fig. 4), which allows for dynamically extrapolating the deep-layer feature output through combining multi-step historical caches based on the shallow-layer probe feature trajectory.

In light of these observations, we introduce **DiCache**, a novel plug-and-play adaptive caching strategy for accelerating diffusion models at runtime, as depicted in Fig. 1 (b). Specifically, DiCache is composed of two principal components: First, *Online Probe Profiling Scheme* is proposed to harness shallow-layer features of the diffusion model to estimate the caching error in real-time, thereby more accurately tailoring the cache schedule for each sample. Second, *Dynamic Cache Trajectory Alignment* is introduced to adaptively approximate the deep-layer feature output from multi-step caches based on the probe feature trajectory, which further elevates visual quality. By integrating the above two techniques, DiCache intrinsically answers when and how to cache within a unified framework.

Our contributions can be summarized as follows: (1) **Shallow-Layer Probe Paradigm**: We introduce an innovative probe-based approach that leverages signals from shallow model layers to predict the caching error and effectively utilize multi-step caches. (2) **DiCache**: We present DiCache, a novel caching strategy that employs online shallow-layer probes to achieve more accurate caching schedules and superior multi-step cache utilization. (3) **Superior Performance**: Comprehensive experiments demonstrate that DiCache consistently delivers higher efficiency and enhanced visual fidelity compared with existing state-of-the-art methods on leading diffusion models including WAN 2.1 (Wan et al., 2025), HunyuanVideo (Kong et al., 2024), and Flux (Labs, 2024).

## 2  RELATED WORK

**Diffusion Model**. Diffusion models (Song & Ermon, 2019; Sohl-Dickstein et al., 2015; Ho et al., 2020; Dhariwal & Nichol, 2021) have demonstrated exceptional capability to synthesize content with superb quality and rich diversity. Early pioneers like Stable Diffusion (Rombach et al., 2022) and SDXL (Podell et al., 2023) achieved high-fidelity image generation with the U-Net (Ronneberger et al., 2015) architecture, which also inspired subsequent video diffusion models (Guo et al., 2023; Chen et al., 2024a; Blattmann et al., 2023). Despite its success, the U-Net architecture possesses limited scalability, hindering the training and deployment of large-scale models with improved performance. To overcome this constraint, recent diffusion models (Esser et al., 2024; Labs, 2024; Chen et al., 2023; Peebles & Xie, 2023) have widely adopted diffusion transformers (DiT) as their backbones for better scalability and flexibility, delivering superior outcomes across various domains (Wu et al., 2025; Wan et al., 2025; Kong et al., 2024; Yang et al., 2024).

**Diffusion Model Acceleration**. Despite the striking achievements drawn by diffusion models, their inference cost increases substantially with model capacity and complexity, posing challenges to practical applications. Accordingly, acceleration techniques (Dao et al., 2022; Li et al., 2024; Liu et al., 2025a) have emerged and gradually become a key research field in the realm of diffusion models, with existing studies generally falling into the following categories: efficient attention (Dao et al., 2022; Zhang et al., 2025b), sparse attention (Xi et al., 2025; Ren et al., 2025), model distillation (Meng et al., 2023; Sauer et al., 2024; Wang et al., 2023), model quantization (Li et al., 2024), improved SDE or ODE solvers (Lu et al., 2022), and caching-based feature reuse strategies (Liu et al., 2025a;b; Zhou et al., 2025). Among them, caching-based methods (Liu et al., 2025a; Zhao et al., 2024) have recently gained increasing attention due to their lightweight nature. For instance, DeepCache (Xu et al., 2018) and Faster Diffusion (Li et al., 2023) reuse the U-Net feature across timesteps to reduce computation redundancy. FORA (Selvaraju et al., 2024) and $\Delta$-DiT (Chen et al., 2024b) extend this idea to transformer-based backbones. PAB (Zhao et al., 2024) broadcasts attention features to subsequent steps in a pyramid style based on different block characteristics. AdaCache (Kahatapitiya et al., 2024) dynamically modifies residual reuse strategies based on the content complexity. FasterCache (Lv et al., 2024) proposes to cache for both the conditional branch and unconditional branch of Classifier-Free Guidance (CFG) (Ho & Salimans, 2022). TeaCache (Liu et al., 2025a) leverages a calibrated polynomial estimator to predict output changes from input differences. EasyCache (Zhou et al., 2025) replaces this polynomial function with an empirical transformation rate law. TaylorSeer (Liu et al., 2025b) suggests combining multi-step cached features in a Taylor-expansion-like manner. Unlike existing methods that rely on predefined empirical laws or dataset-level priors, we enable the diffusion model to autonomously determine the caching timings and adaptively utilize multi-step caches according to an online probe at runtime, thereby better adapting to the highly dynamic diffusion process and the diverse sample distribution.

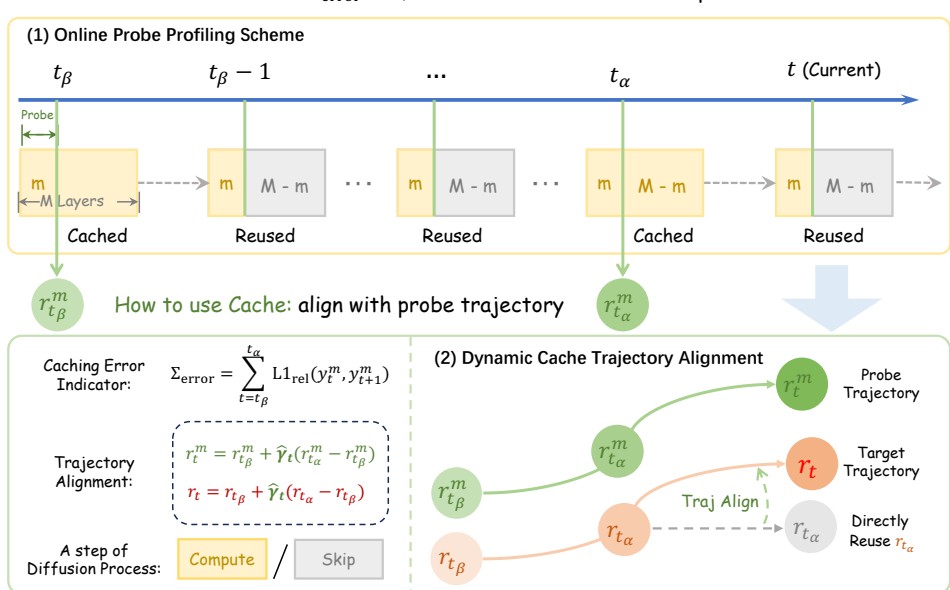

Figure 2: **Overview of DiCache**. The proposed DiCache consists of Online Probe Profiling Scheme and Dynamic Cache Trajectory Alignment. The former dynamically determines the caching timing with an online shallow-layer probe at runtime, while the latter combines multi-step caches based on the probe feature trajectory to adaptively approximate the feature at the current timestep.

## 3 METHOD

### 3.1 PRELIMINARY

**Flow Matching**. Flow Matching (Lipman et al., 2022) and Rectified Flow (Liu et al., 2022) are designed to simplify the construction of Ordinary Differential Equation (ODE) models by delineating a linear mapping between disparate probability distributions. Given samples $x_1$ drawn from the noise distribution $\pi_1$ and $x_0$ from the clean data distribution $\pi_0$, the transition from $x_1$ to $x_0$ is modeled as a linear trajectory in the direction of the vector $(x_0 - x_1)$, with the intermediate state at timestep $t$ ($t \in [0, 1]$) denoted as $x_t = tx_1 + (1 - t)x_0$. Hence, the ODE governing $x_t$ can be formulated as $dx_t = (x_0 - x_1)dt$. Since $x_0$ is unknown during denoising, a learned velocity field $v_\theta(x_t, t, c)$ is employed to approximate the direction $(x_0 - x_1)$, thereby constructing a neural ODE model:

$$d\hat{x}_t = v_\theta(x_t, t, c)dt, \tag{1}$$

where $c$ stands for extra input conditions such as textual or image prompts.

**Diffusion Transformer**. The Diffusion Transformer (DiT) (Peebles & Xie, 2023; Esser et al., 2024) architecture adopts a tiered layout with $M$ cascaded blocks, in which self-attention (SA), cross-attention (CA), and a multilayer perceptron (MLP) are integrated as a unified module. Let $\mathcal{B}_i$ denote the $i$-th DiT block ($i \in [1, M]$), $y_t^i$ the output of the $i$-th DiT block at timestep $t$, the transformation between $y_t^i$ and model input latent $x_t$ can be formulated as:

$$y_t^i = \mathcal{B}_i \circ \mathcal{B}_{i-1} \circ \cdots \circ \mathcal{B}_2 \circ \mathcal{B}_1(x_t, t, c), \tag{2}$$

in which $y_t^M$ represents the final model output at timestep $t$, i.e. the predicted velocity $v_\theta(x_t, t, c)$.

**Residual in DiT**. The residual between the output of the $i$-th DiT block $y_t^i$ and the initial model input latent $x_t$ at timestep $t$ can be expressed as:

$$r_t^i = y_t^i - x_t, \tag{3}$$

in which $r_t^M = y_t^M - x_t = v_\theta(x_t, t, c) - x_t$ represents the difference between the model's predicted velocity and its input, i.e. the residual of the entire diffusion transformer. Since the residual effectively captures the evolution of features across DiT blocks, we choose to cache for model residuals.

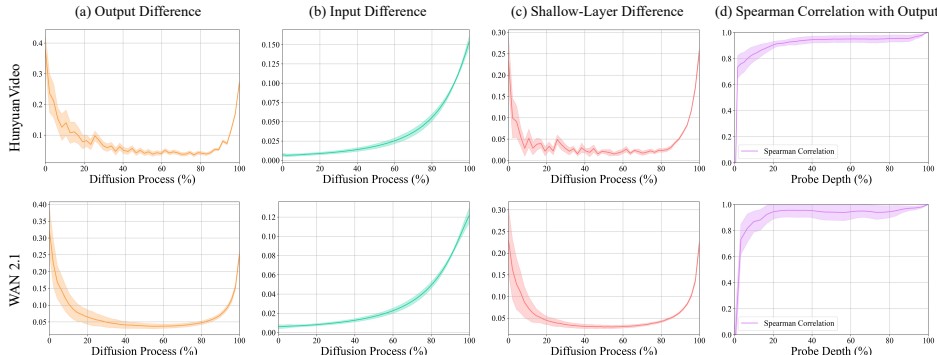

Figure 3: **Variation of feature differences between consecutive timesteps (Mean & Variance).** (a) Output differences $\mathrm{L1_{rel}}(y_t, y_{t+1})$. It exhibits large variances, indicating sample-specific nature. (b) Input differences $\mathrm{L1_{rel}}(x_t, x_{t+1})$. It increases monotonically with timesteps, failing to capture the variations in output differences. (c) Shallow-layer feature differences $\mathrm{L1_{rel}}(y_t^m, y_{t+1}^m)$ (5-th layer in this figure). It exhibits a strong correlation with the variations in output differences. (d) Spearman correlation coefficient between $\mathrm{L1_{rel}}(y_t^m, y_{t+1}^m)$ and $\mathrm{L1_{rel}}(y_t, y_{t+1})$. They already exhibit a high correlation coefficient (around 0.8) with a shallow probe depth (1~3 layers).

## 3.2 DiCache

Instead of resorting to data-driven or empirical metrics, DiCache employs an online shallow-layer probe to efficiently obtain an indicator of diffusion dynamics, offering guidance for both caching timing determination and multi-step cache utilization. Specifically, we propose *Online Probe Profiling Scheme* to address *"When to cache"* and *Dynamic Cache Trajectory Alignment* to answer *"How to use Cache"*, respectively. The overall framework of DiCache is illustrated in Fig. 2.

**Online Probe Profiling Scheme**. We begin by analyzing the caching error. Given that the optimal moment to reuse cached features depends on the evolution of model outputs over timesteps (Liu et al., 2025a), the ideal caching error $\epsilon_{t,t+1}$ can be defined as the difference between model outputs at adjacent timesteps $t$ and $t+1$:

$$\epsilon_{t,t+1} = \mathrm{L1_{rel}}(y_t, y_{t+1}) = \frac{||y_t - y_{t+1}||_1}{||y_{t+1}||_1}, \tag{4}$$

where $y_t$ denotes the model output at timestep $t$ and the relative L1 distance ($\mathrm{L1_{rel}}(\cdot)$) is adopted to measure feature differences. Once $\epsilon_{t,t+1}$ is known, it is able to determine whether the cache should be used at timestep $t$. However, previous studies seek to anticipate these output variations using offline priors (Liu et al., 2025a) or empirical heuristics (Zhou et al., 2025), overlooking the specificity of individual samples, thus resulting in suboptimal generalizability on outlier instances. Unlike existing works, we discover that: **for a specific sample, the variation patterns of its shallow-layer feature differences remain consistent with those of its deep-layer feature, demonstrating that an efficient indicator for caching error can be obtained without delving deep into the model.** Consider a diffusion model with $M$ blocks/layers, let $x_t$ denote the input and $y_t^m$ the output of the $m$-th layer ($m \in [1, M]$, $y_t^M = y_t$, i.e. the final model output) at timestep $t$, we demonstrate that $\mathrm{L1_{rel}}(y_t^m, y_{t+1}^m)$ can reflect the dynamics of $\mathrm{L1_{rel}}(y_t^M, y_{t+1}^M)$ with $m << M$. As shown in Fig. 3 (a)-(c), while $\mathrm{L1_{rel}}(x_t, x_{t+1})$ fails to capture the variation pattern of $\mathrm{L1_{rel}}(y_t^M, y_{t+1}^M)$, $\mathrm{L1_{rel}}(y_t^m, y_{t+1}^m)|_{m=5}$ effectively characterizes it with only $m = 5$. Moreover, as depicted in Fig. 3 (d), the Spearman correlation coefficient between $\mathrm{L1_{rel}}(y_t^m, y_{t+1}^m)$ and $\mathrm{L1_{rel}}(y_t^M, y_{t+1}^M)$ reaches a significant level (around 0.8) even when $m$ is very small ($m \in [1, 3]$).

Motivated by this insight, $\mathrm{L1_{rel}}(y_t^m, y_{t+1}^m)$ can be utilized as an estimated caching error with a $m$-th layer online probe:

$$\hat{\epsilon}_{t,t+1} = \mathrm{L1_{rel}}(y_t^m, y_{t+1}^m) = \frac{||y_t^m - y_{t+1}^m||_1}{||y_{t+1}^m||_1}. \tag{5}$$

Based on the above analysis, we propose an Online Probe Profiling Scheme to efficiently and accurately determine caching timings at inference time, as detailed in Algorithm 1. Instead of relying

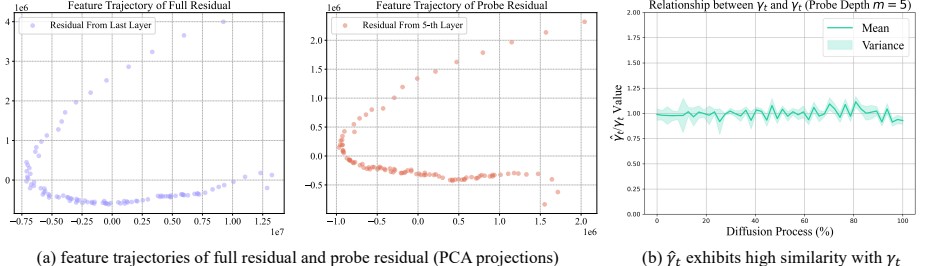

(a) feature trajectories of full residual and probe residual (PCA projections)  (b) $\hat{\gamma}_t$ exhibits high similarity with $\gamma_t$

Figure 4: **Observation and analysis regarding Dynamic Cache Trajectory Alignment.**

on pre-trained estimators or other empirical metrics, we only infer the first $m$ ($m << M$) layers of the model as a shallow-layer probe at each timestep to obtain a proxy of $\text{L1}_{\text{rel}}(y_t^M, y_{t+1}^M)$ and compute estimated caching error $\hat{\epsilon}_{t,t+1}$ via Eq. 5. An accumulated caching error is maintained across timesteps as a metric for caching error tolerance:

$$\sum_{t=t_1}^{t_2+1} \hat{\epsilon}_{t,t+1} \leq \delta < \sum_{t=t_1}^{t_2} \hat{\epsilon}_{t,t+1}, \tag{6}$$

in which $\delta$ is a user-specified threshold. Specifically, after computing and caching the model residual at timestep $t_1$, we accumulate the estimated caching error $\hat{\epsilon}_{t,t+1}$ and reuse the cached residual for each subsequent timestep. When the accumulated error exceeds the threshold $\delta$ at timestep $t_2$ ($t_2 < t_1$), the residual is recomputed to refresh the cache and the accumulated caching error is reset to zero. It is noteworthy that our strategy inherently supports **resuming computation from the probed layer**, thereby incurring no additional cost at timesteps requiring recomputation.

---

**Algorithm 1** Online Probe Profiling Scheme

---
1: **Inputs:** Model input $x_T$, Sampling steps $T$, DiT blocks $\mathcal{B}_i$ ($i \in [1, M]$), Extra conditions $c$
2: **Parameters:** Reuse threshold $\delta$, Probe depth $m$ ($m << M$)
3: Initialize accumulated estimated caching error $\Sigma_{\text{error}} \leftarrow 0$, cached residual $R \leftarrow$ `None`
4: **for** $t = T$ to $0$ **do**
5:     **if** $t == T$ **then**
6:         $y_t \leftarrow \mathcal{B}_M \circ \cdots \circ \mathcal{B}_1(x_t, t, c)$ # Calculate for the first step
7:         $R \leftarrow y_t - x_t$ # Initialize cache from `None`
8:     **else**
9:         $y_t^m \leftarrow \mathcal{B}_m \circ \cdots \circ \mathcal{B}_1(x_t, t, c)$ # Probe the first $m$ layers
10:         $\Sigma_{\text{error}} \leftarrow \Sigma_{\text{error}} + \text{L1}_{\text{rel}}(y_t^m, y_{t+1}^m)$ # Accumulate caching error
11:         **if** $\Sigma_{\text{error}} \leq \delta$ **then**
12:             $y_t \leftarrow x_t + R$ # Reuse cached residual
13:         **else**
14:             $y_t \leftarrow \mathcal{B}_M \circ \cdots \circ \mathcal{B}_{m+1}(y_t^m)$ # Resume from the probed layer
15:             $R \leftarrow y_t - x_t, \Sigma_{\text{error}} \leftarrow 0$ # Refresh cache and error accumulator
16:         **end if**
17:     **end if**
18: **end for**

---

**Dynamic Cache Trajectory Alignment**. While Online Probe Profiling Scheme resolves *"When to Cache"*, the generation quality is still limited by insufficient cache utilization. To this end, we further propose a simple yet effective Dynamic Cache Trajectory Alignment (DCTA) strategy to answer *"How to use cache"*. Previous works (Lv et al., 2024; Liu et al., 2025b) usually employ fixed handcrafted rules such as Taylor expansion for utilizing multi-step caches, lacking adaptability and risking deviation from original results, as depicted in Fig. 1 (c)(3). In this work, we discover that **features from different model layers exhibit similar trajectories**. As shown in Fig. 4 (a), the shallow-layer probe residual from the $m$-th layer $r_t^m$ ($m = 5$ here) and the residual of the entire model $r_t$ display analogous dynamic trends. Since the residual represents the transformation direction from the latent space input to the velocity space output, for a well-trained DiT, these directions are generally aligned across different blocks. Therefore, a reliable signal can be derived from the shallow-layer probe to guide the utilization of multi-step cached residuals for a better approximation of the current residual.

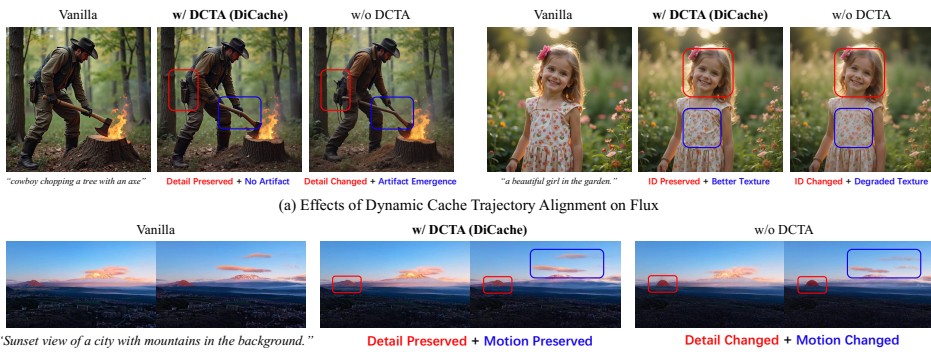

(a) Effects of Dynamic Cache Trajectory Alignment on Flux

(b) Effects of Dynamic Cache Trajectory Alignment on WAN 2.1

Figure 5: **Effects of Dynamic Cache Trajectory Alignment (DCTA). Best viewed zoomed in.**

Assuming the two most recent recomputation timesteps are $t_\alpha$ and $t_\beta$ with $t_\alpha < t_\beta$, and they cache model residuals $r_{t_\alpha}$ and $r_{t_\beta}$, respectively. We compute the current residual in the following form:

$$r_t = r_{t_\beta} + \gamma_t(r_{t_\alpha} - r_{t_\beta}), \tag{7}$$

where $\gamma_t$ stands for the residual trajectory parameter at timestep $t$. It characterizes the dynamic trend of the residual's feature trajectory at each timestep. By emphasizing such trends of residual changes across timesteps, Eq. 7 enables a more precise approximation of the current residual compared to directly using the most recent cache $r_{t_\alpha}$. More discussion on the design of Eq. 7 can be found in Section F in the appendix. Nevertheless, since the diffusion process is unpredictable, the optimal value of $\gamma_t$ is unknown before the model is inferred at timestep $t$. Here, we propose to leverage the online probe feature trajectory to dynamically estimate $\gamma_t$ value at each timestep. Specifically, after computing the probe feature $y_t^m$, its corresponding probe residual is calculated as:

$$r_t^m = y_t^m - x_t. \tag{8}$$

Since the probe is computed at each timestep, the ground-truth values of $r_t^m$, $r_{t_\alpha}^m$ and $r_{t_\beta}^m$ are already available. Therefore, the ideal feature trajectory can be established among them:

$$r_t^m = r_{t_\beta}^m + \hat{\gamma}_t(r_{t_\alpha}^m - r_{t_\beta}^m), \tag{9}$$

in which the probe residual trajectory parameter $\hat{\gamma}_t$ can be directly solved by:

$$\hat{\gamma}_t = \frac{\text{L1}_{\text{rel}}(r_t^m, r_{t_\beta}^m)}{\text{L1}_{\text{rel}}(r_{t_\alpha}^m, r_{t_\beta}^m)}. \tag{10}$$

Given the similarity between the probe residual trajectory and the full residual trajectory, $\hat{\gamma}_t$ exhibits high consistency with $\gamma_t$ even under small $m$ values, as shown in Fig. 4 (b). Consequently, $\hat{\gamma}_t$ can be utilized as an efficient substitute for $\gamma_t$ in Eq. 7:

$$r_t = r_{t_\beta} + \hat{\gamma}_t(r_{t_\alpha} - r_{t_\beta}) = r_{t_\beta} + \frac{\text{L1}_{\text{rel}}(r_t^m, r_{t_\beta}^m)}{\text{L1}_{\text{rel}}(r_{t_\alpha}^m, r_{t_\beta}^m)}(r_{t_\alpha} - r_{t_\beta}), \tag{11}$$

which allows for adaptive adjustment of the trajectory parameter, thereby more accurately estimating the full residual with multi-step caches and facilitating better visual quality, as illustrated in Fig. 5.

## 4 EXPERIMENTS

### 4.1 IMPLEMENTATION DETAILS

**Experimental Settings**. We conduct our experiments on three leading DiT-based diffusion models: WAN 2.1-1.3B (Wan et al., 2025) and HunyuanVideo (Kong et al., 2024) for video generation, and Flux.1.0-dev (Labs, 2024) for image generation. For hyperparameters, we set probe depth $m = 1$ for all three models, while reuse threshold is set as $\delta = 0.2$ for WAN 2.1, $\delta = 0.1$ for HunyuanVideo, and $\delta = 0.4$ for Flux, respectively. If not specified, experiments are conducted on a single NVIDIA A800 80GB GPU. More implementation details can be found in Section B in the appendix.

**Baselines**. The compared methods encompass (i) two naive acceleration strategies (directly reducing the number of timesteps by 50% and uniform caching with different static cache intervals) as well as (ii) various state-of-the-art caching-based methods, including TeaCache (Liu et al., 2025a), TaylorSeer (Liu et al., 2025b), EasyCache (Zhou et al., 2025) and ToCa (Zou et al., 2024).

**Evaluation Metrics**. We collect 300 high-quality text prompts from VideoDPO (Liu et al., 2025c) and VideoFeedback (He et al., 2024) datasets (polished by GPT-4o (Hurst et al., 2024)) for video generation and 1K captions sourced from the LAION-5B (Schuhmann et al., 2022) dataset for image generation. Following previous works (Liu et al., 2025a; Zhou et al., 2025), speedup ratio and latency are reported to measure inference efficiency, while LPIPS (Zhang et al., 2018), PSNR, and SSIM regarding original results are adopted to assess visual quality.

## 4.2 COMPARISON TO EXISTING METHODS

**Quantitative Evaluation**. The quantitative assessments are presented in Tab. 1. We begin by comparing DiCache with two basic cache strategies: direct step reduction and uniform caching with a static caching interval $I$. Despite their simplicity, they fail to adapt to the dynamic diffusion process, resulting in considerable quality loss (about 30% SSIM reductions and 40% PSNR loss compared to DiCache on WAN 2.1). While TaylorSeer achieves an impressive $3.43\times$ speedup on WAN 2.1, its generation results deviate significantly from the original with an LPIPS of $0.5214$. Moreover, its long-range feature forecast mechanism imposes a severe burden on GPU memory, even encountering out-of-memory issues on HunyuanVideo when using a single A800 GPU. TeaCache overly relies on data priors and is prone to overfitting to training prompts, leading to unstable performance on unseen cases. Though TeaCache-slow maintains relatively acceptable quality at a low acceleration rate, a notable performance drop occurs once pushing acceleration further to TeaCache-fast (with an LPIPS of $0.2898$, a SSIM of $0.8015$ and a PSNR of $22.01$ on HunyuanVideo). EasyCache yields suboptimal efficiency across various models due to its inability to precisely capture the diffusion dynamics with an empirical transformation rate metric. In contrast, our proposed DiCache dynamically determines its caching timings and effectively utilizes multi-step caches based on online probes, achieving a unification of rapid inference speed and high visual fidelity (e.g., LPIPS $0.1492$, SSIM $0.9396$, PSNR $32.79$ and $2.34\times$ speedup on HunyuanVideo). On the image generation backbone Flux.1.0-dev, the benefits of DiCache become even more pronounced, with markedly superior performance in efficiency and quality across all evaluated baselines. In addition, DiCache can be integrated with other acceleration methods to offer even higher speed (detailed in Section 4.3).

Table 1: **Quantitative comparison with baselines.** The best result is highlighted in **bold**, while the second-best result is underlined. "OOM" indicates CUDA out of memory on the A800 80GB GPU.

| Model | Method | LPIPS ↓ | SSIM ↑ | PSNR ↑ | Speedup ↑ | Latency ($sec$) ↓ |
|---|---|---|---|---|---|---|
| WAN 2.1 | Vanilla (100% steps) | - | - | - | $1.00\times$ | 192.47 |
| | Vanilla (50% steps) | 0.4143 | 0.6304 | 16.19 | $1.83\times$ | 105.25 |
| | Uniform Cache ($I=2$) | 0.4740 | 0.5927 | 15.16 | $2.39\times$ | 80.44 |
| | TeaCache-slow (Liu et al., 2025a) | 0.1939 | 0.8374 | 22.60 | $1.82\times$ | 105.83 |
| | TeaCache-fast (Liu et al., 2025a) | 0.2161 | 0.8226 | 20.97 | $2.20\times$ | 87.58 |
| | TaylorSeer (Liu et al., 2025b) | 0.5214 | 0.5485 | 14.32 | $\mathbf{3.43\times}$ | **56.05** |
| | EasyCache (Zhou et al., 2025) | 0.2013 | 0.8562 | 24.80 | $2.21\times$ | 86.96 |
| | **DiCache (Ours)** | **0.1734** | **0.8885** | **26.45** | $2.45\times$ | 78.42 |
| HunyuanVideo | Vanilla (100% steps) | - | - | - | $1.00\times$ | 1186.32 |
| | Vanilla (50% steps) | 0.4138 | 0.7134 | 17.90 | $1.98\times$ | 599.65 |
| | Uniform Cache ($I=2$) | 0.4111 | 0.7132 | 17.92 | $1.91\times$ | 622.45 |
| | TeaCache-slow (Liu et al., 2025a) | 0.2762 | 0.8114 | 22.41 | $1.65\times$ | 717.21 |
| | TeaCache-fast (Liu et al., 2025a) | 0.2898 | 0.8015 | 22.01 | $2.20\times$ | 538.49 |
| | TaylorSeer (Liu et al., 2025b) | OOM | OOM | OOM | OOM | OOM |
| | EasyCache (Zhou et al., 2025) | 0.1558 | 0.9270 | 30.71 | $2.12\times$ | 558.71 |
| | **DiCache (Ours)** | **0.1492** | **0.9396** | **32.79** | **$2.34\times$** | **507.24** |
| Flux | Vanilla (100% steps) | - | - | - | $1.00\times$ | 15.11 |
| | Vanilla (50% steps) | 0.3679 | 0.7370 | 18.04 | $1.98\times$ | 7.63 |
| | Uniform Cache ($I=3$) | 0.4034 | 0.7013 | 17.09 | $2.55\times$ | 5.92 |
| | TeaCache-slow (Liu et al., 2025a) | 0.2810 | 0.8036 | 21.81 | $1.71\times$ | 8.86 |
| | TeaCache-fast (Liu et al., 2025a) | 0.4053 | 0.7219 | 18.01 | $2.82\times$ | 5.36 |
| | ToCa (Zou et al., 2024) | 0.3837 | 0.7592 | 20.60 | $1.52\times$ | 9.92 |
| | TaylorSeer (Liu et al., 2025b) | 0.4709 | 0.6721 | 16.63 | $3.13\times$ | 4.83 |
| | EasyCache (Zhou et al., 2025) | 0.3049 | 0.7527 | 19.75 | $2.49\times$ | 6.06 |
| | **DiCache (Ours)** | **0.2704** | **0.8211** | **22.39** | $3.22\times$ | **4.69** |

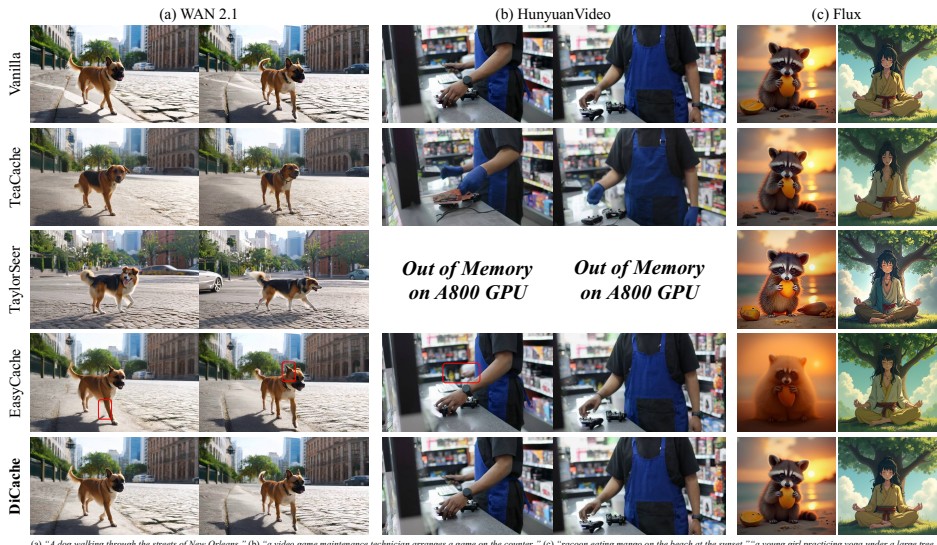

Figure 6: **Qualitative comparison with previous acceleration methods. Best viewed zoomed in.**

**Qualitative Comparison**. Fig. 6 presents the visual comparison between the proposed DiCache and other baselines. While TeaCache achieves a notable speedup, the visual quality suffers a significant degradation. For instance, compared with the vanilla results, the textures of the dog and the road surface generated by TeaCache exhibit a marked loss. TaylorSeer employs a Taylor expansion-based feature forecast mechanism to explicitly enhance texture details. However, it often manifests low similarity to the original results and exhibits abnormal color contrast (especially with Flux). Easy-Cache adopts a predefined cache law, suffering from inaccurate caching timing determination and insufficient cache utilization. The results produced by EasyCache deviate from the original results in critical details such as the dog's ears and human hands. Moreover, such a heuristic rule sometimes leads to generation collapse, resulting in degraded outputs (e.g., the "raccoon" case generated by EasyCache with Flux). In comparison, the proposed DiCache consistently outperforms the baselines in terms of both visual quality and similarity to the original results across diverse scenarios and generation backbones. More visual comparison results are presented in Section C in the appendix.

## 4.3 COMPATIBILITY WITH OTHER ACCELERATION TECHNIQUES

We analyze the proposed DiCache's compatibility with other acceleration techniques like sparse attention and model distillation to demonstrate its flexibility and versatility. Experiments in this section are conducted on a single NVIDIA H200 140GB GPU.

**Compatibility with Sparse Attention**. We integrate DiCache with the advanced sparse attention method Sparse VideoGen (SVG) (Xi et al., 2025) on HunyuanVideo ($720 \times 1280$ resolution, 129 frames). As depicted in Fig. 14 in the appendix, by combining DiCache with SVG, we achieve a $3.08\times$ speedup with negligible loss in visual quality. Moreover, as shown in Tab. 2, the incorporation of DiCache into SVG leads to a significant improvement in efficiency compared to using SVG alone (from $1.67\times$ speedup to $3.08\times$ speedup), without a noticeable degradation in quantitative metrics.

Table 2: **DiCache's compatibility with the sparse attention method Sparse VideoGen (SVG).**

| Model | Method | LPIPS ↓ | SSIM ↑ | PSNR ↑ | Speedup ↑ | Latency (sec) ↓ |
|---|---|---|---|---|---|---|
| | Vanilla | - | - | - | 1.00× | 1747.30 |
| HunyuanVideo | + SVG (Xi et al., 2025) | **0.1760** | **0.9034** | **28.18** | 1.67× | 1047.63 |
| | **+ SVG (Xi et al., 2025) + DiCache** | 0.2051 | 0.8882 | 27.20 | **3.08×** | **567.45** |

**Compatibility with Model Distillation**. We combine DiCache with the distilled version of WAN 2.1-14B (by AccVideo (Zhang et al., 2025a), with 10 sampling steps), achieving a notable accelera-

tion without compromising visual quality. The quantitative evaluation is presented in Tab. 3, while the qualitative results are shown in Fig. 16 in the appendix.

Table 3: **DiCache's compatibility with the model distillation method AccVideo.**

| Model | Method | LPIPS ↓ | SSIM ↑ | PSNR ↑ | Speedup ↑ | Latency ($sec$) ↓ |
|---|---|---|---|---|---|---|
| WAN 2.1-14B | AccVideo (Zhang et al., 2025a) | - | - | - | 1.00× | 48.24 |
| | **AccVideo (Zhang et al., 2025a)+ DiCache** | **0.2737** | **0.7432** | **19.72** | **1.56×** | **30.98** |

## 4.4 ABLATION AND ANALYSIS

We conducted ablation studies on the video generation model HunyuanVideo, as shown in Tab. 4.

**Choice of Probe Depth** $m$. $m$ determines the number of DiT blocks/layers that are probed at each sampling step. Empirically, a larger $m$ leads to more accurate cache timing decisions but also incurs a higher probe time cost. However, as illustrated in Fig. 3 (d), $\text{L1}_{\text{rel}}(y_t^m, y_{t+1}^m)$ and $\text{L1}_{\text{rel}}(y_t^M, y_{t+1}^M)$ already exhibit a strong correlation when $m$ is very small ($m \in [1, 3]$). Therefore, $m = 1$ is chosen to offer higher efficiency. Ablation results on probe depth $m$ are presented in Tab. 4 (a).

**Choice of Reuse Threshold** $\delta$. The value of $\delta$ stands for the tolerance level for caching errors (i.e., the trade-off between quality and efficiency). As depicted in Tab. 4 (b), adopting a smaller $\delta$ yields superior visual quality, yet concurrently results in increased latency, as the model is inferred at more timesteps. Conversely, a larger $\delta$ facilitates higher inference speed but entails a certain degree of quality loss. $\delta = 0.1$ is chosen in our experiments to strike a balance between quality and efficiency.

**Effects of Dynamic Cache Trajectory Alignment (DCTA)**. Dynamic Cache Trajectory Alignment aims at better approximating features at the current timestep through combining multi-step caches based on shallow-layer probe feature trajectory. As shown in Fig. 5 and Tab. 4 (c), Dynamic Cache Trajectory Alignment leads to improved visual quality and similarity regarding original results. Additional visual ablation results of DCTA are presented in Fig. 15 in the appendix.

Table 4: **Ablation experiments on DiCache components and hyperparameters.**

| Components | Values & Choices | LPIPS ↓ | SSIM ↑ | PSNR ↑ | Speedup ↑ | Latency ($sec$) ↓ |
|---|---|---|---|---|---|---|
| (a) Probe Depth $m$ | $m = 5$ | **0.1367** | **0.9495** | **33.47** | 2.10× | 563.68 |
| | $m = 3$ | 0.1397 | 0.9472 | 33.30 | 2.20× | 539.83 |
| | **$m = 1$** | 0.1492 | 0.9396 | 32.79 | **2.34×** | **507.24** |
| (b) Reuse Threshold $\delta$ | $\delta = 0.20$ | 0.1886 | 0.8980 | 29.81 | **2.90×** | **408.62** |
| | $\delta = 0.15$ | 0.1665 | 0.9131 | 30.92 | 2.62× | 452.33 |
| | $\delta = \mathbf{0.10}$ | 0.1492 | 0.9396 | 32.79 | 2.34× | 507.24 |
| | $\delta = 0.08$ | 0.1305 | 0.9433 | 34.03 | 2.10× | 564.73 |
| | $\delta = 0.05$ | **0.1047** | **0.9584** | **35.45** | 1.76× | 672.20 |
| (c) DCTA | w/o | 0.1517 | 0.9314 | 31.98 | **2.34×** | **506.82** |
| | **w/** | **0.1492** | **0.9396** | **32.79** | **2.34×** | 507.24 |

## 5 CONCLUSION

We present DiCache, a novel training-free adaptive caching strategy for accelerating diffusion models at runtime. DiCache consists of Online Probe Profiling Scheme and Dynamic Cache Trajectory Alignment, the former dynamically determines caching timings based on a shallow-layer online probe, while the latter adaptively utilizes multi-step caches according to the probe feature trajectory. Extensive experiments validate the effectiveness of DiCache on various leading diffusion models.

## 6 ACKNOWLEDGEMENT

This project is funded in part by Shanghai Artificial Intelligence Laboratory, Shanghai Innovation Institute, the Centre for Perceptual and Interactive Intelligence (CPII) Ltd under the Innovation and Technology Commission (ITC)'s InnoHK. Dahua Lin is a PI of CPII under the InnoHK.

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

## A APPENDIX

In the appendix, we present additional implementation details (Section B), additional qualitative results (Section C), text prompts used in both the main paper and appendix (Section D), more analysis on the choice of reuse threshold (Section E), more discussion on Dynamic Cache Trajectory Alignment (Section F), the limitations of our method (Section G), the ethical statement (Section H), the reproducibility statement (Section I), the declaration on LLM usage (Section J), as well as the description of supplementary material K, as a supplement to the main paper.

## B ADDITIONAL IMPLEMENTATION DETAILS

The detailed configurations adopted by each model in the main experiments are as follows: ($832 \times 480$ resolution, 81 frames, 50 sampling steps) for WAN 2.1, ($544 \times 960$ resolution, 129 frames, 50 sampling steps) for HunyuanVideo, and ($1024 \times 1024$ resolution, 30 sampling steps) for Flux.

## C ADDITIONAL QUALITATIVE RESULTS

In this section, we present more visual comparison results between DiCache and existing caching-based methods in Fig. 7, Fig. 8, Fig. 9, Fig. 10, Fig. 11 and Fig. 12, along with accelerated results using the same prompts and different random seeds ($0/1/2$) presented in Fig. 13.

## D TEXT PROMPTS

All text prompts used to generate images or videos in this paper are listed in Tab. 8.

## E MORE ANALYSIS ON THE CHOICE OF REUSE THRESHOLD

Admittedly, the recommended value of the reuse threshold $\delta$ varies across models, but the differences are relatively minor. These variations arise from the differences among DiT models with different training data and architectures (e.g., Flux is a T2I model with 57 layers and 12B parameters, while WAN 2.1-1.3B is a T2V model with 30 layers and 1.3B parameters). A potential way to calibrate $\delta$ for a new model is to perform a simple one-dimensional sweep over a commonly used threshold range (e.g., a $\delta \in [0.1, 0.5]$ is recommended since it has shown effectiveness across most tested models) using a step size of $0.05$ or $0.1$. Furthermore, once the value of $\delta$ is determined, it exhibits strong stability across different samples within the same model. Specifically, we sample 1K diverse prompts (the same evaluation set as the main paper) from the LAION-5B dataset and use Flux for image generation. The results demonstrate that DiCache not only maintains the lowest LPIPS but also exhibits the lowest LPIPS variance compared to state-of-the-art methods (as shown in Tab. 5), indicating its stable performance across diverse samples and strong generalization capability.

Table 5: **Mean and Variance of LPIPS on Flux**

| Method | Mean (LPIPS)↓ | Variance (LPIPS)↓ |
|---|---|---|
| TeaCache (Liu et al., 2025a) | 0.4053 | 0.010 |
| TaylorSeer (Liu et al., 2025b) | 0.4709 | 0.007 |
| EasyCache (Zhou et al., 2025) | 0.3049 | 0.008 |
| **DiCache (Ours)** | **0.2704** | **0.004** |

## F MORE DISCUSSION ON DYNAMIC CACHE TRAJECTORY ALIGNMENT

In this section, we provide a deeper analysis on the design of Dynamic Cache Trajectory Alignment. As described in the main paper, considering the two most recent cached model residuals $r_{t_\alpha}$ and $r_{t_\beta}$ with corresponding timesteps $t_\alpha < t_\beta$, we estimate the current model residual $r_t$ as:

$$r_t = r_{t_\beta} + \gamma_t(r_{t_\alpha} - r_{t_\beta}), \quad (12)$$

in which $\gamma_t$ stands for the residual trajectory parameter which we dynamically obtained from Eq. 10. Actually, Eq. 12 is in the form of a first-order approximation (with respect to directly using $r_t = r_{t_\alpha}$,

i.e. a zero-order approximation adopted by previous works (Liu et al., 2025a; Zhou et al., 2025)). Since the model residual $r_t$ is a continuously timestep-varying feature with a stable trajectory (as shown in Fig. 4 (a)), utilizing multi-order historical caches for estimation can effectively improve the accuracy of the approximation, as evidenced by the visual comparison results in Fig. 5 and the quantitative ablation results in Tab. 4 (c). Given the effectiveness of Eq. 12, a natural extension is to scale this approach to higher orders. Assuming the recomputation timesteps for a diffusion process form a sequence $t_0, t_1, t_2, \cdots, t_k$ ($t_0 < t_1 < t_2 < \cdots < t_k$), the $k$-th order Dynamic Cache Trajectory Alignment can be formulated as:

$$r_t = r_{t_0} + \sum_{i=1}^{k} \gamma_t \cdot \xi^{i-1} \cdot (r_{t_0} - r_{t_i}),$$ (13)

in which $r_{t_i}$ denotes the cached model residual at timestep $t_i$ ($i \in [0, k]$) and $\xi \in [0, 1]$ is a decay factor that controls the weights of different historical caches in the combination. Similarly, such a relationship can also be established among $k$-th order probe residuals $r_{t_0}^m, r_{t_1}^m, r_{t_2}^m, \cdots, r_{t_k}^m$ with a probe depth $m$:

$$r_t^m = r_{t_0}^m + \sum_{i=1}^{k} \hat{\gamma}_t \cdot \xi^{i-1} \cdot (r_{t_0}^m - r_{t_i}^m),$$ (14)

in which $\hat{\gamma}_t$ represents the probe residual trajectory parameter. Since $r_{t_0}^m, r_{t_1}^m, r_{t_2}^m, \cdots, r_{t_k}^m$ and $r_t^m$ are already computed, $\hat{\gamma}_t$ can be solved by:

$$\hat{\gamma}_t = \frac{\text{L1}_{\text{rel}}(r_t^m, r_{t_0}^m)}{\sum_{i=1}^{k} \xi^{i-1} \cdot \text{L1}_{\text{rel}}(r_{t_0}^m, r_{t_i}^m)}.$$ (15)

Following the main paper, $\hat{\gamma}_t$ is substituted into Eq. 13 to replace $\gamma_t$:

$$r_t = r_{t_0} + \frac{\sum_{i=1}^{k} \xi^{i-1} \cdot \text{L1}_{\text{rel}}(r_t^m, r_{t_0}^m) \cdot (r_{t_0} - r_{t_i})}{\sum_{i=1}^{k} \xi^{i-1} \cdot \text{L1}_{\text{rel}}(r_{t_0}^m, r_{t_i}^m)}.$$ (16)

To evaluate the benefits of increasing the order of multi-step cache utilization, we randomly sampled 500 prompts from the LAION-5B (Schuhmann et al., 2022) dataset and conducted quantitative ablation experiments on Flux ($1024 \times 1024$ resolution). Similar to Section 4.3, experiments in this section are conducted on a single NVIDIA H200 140GB GPU. The decay factor $\xi$ is set as $0.5$. As can be observed in Tab. 6, the benefits of further increasing the cache utilization order $k$ are relatively minor compared to the performance gain from zero-order to first-order approximation. In addition, storing features at more timesteps can impose a greater memory burden, especially for video generation models with a large number of frames (e.g., HunyuanVideo). **Considering both simplicity and efficiency**, we adopt the first-order approximation for Dynamic Cache Trajectory Alignment.

Table 6: **Ablation study on the order of multi-step cache utilization in Dynamic Cache Trajectory Alignment.** The performance gain is measured using the reduction ratio of LPIPS.

| Model | Components | Values & Choices | LPIPS ↓ | SSIM ↑ | PSNR ↑ | Performance Gain (w.r.t. $k = 0$) |
|-------|-----------|------------------|---------|--------|--------|-----------------------------------|
| Flux | Cache Order $k$ | $k = 0$ | 0.2930 | 0.8166 | 21.83 | 0.00% |
| | | $k = 1$ | 0.2691 | 0.8245 | 22.43 | 8.16% (+8.16%) |
| | | $k = 2$ | **0.2686** | 0.8228 | 22.16 | 8.33% (+0.17%) |
| | | $k = 4$ | 0.2710 | **0.8269** | **22.55** | 7.51% (-0.82%) |

## G  LIMITATION AND FUTURE WORKS

Despite the advancements of DiCache in accelerating diffusion models, it faces certain constraints. Specifically, the two principal components of DiCache both rely on the online probe at runtime. Even though we have validated that probing shallow layers is sufficient (e.g., for the 57-layer Flux model, probe only the first layer with a probing rate of $1/57 \approx 0.018$), the current probing paradigm still incurs a certain amount of time cost. Future works can focus on exploring ways to further reduce the probing cost, thereby more efficiently obtaining the output dynamics in the diffusion process.

In addition, we provide a comparison of the probing cost $T_{\text{probe}}$ across different architectures, as well as the ratio of the probing cost to the total inference time with DiCache ($T_{\text{prob}}/T_{\text{all}}$), as shown in Tab. 7. Since DiT models are typically deep while DiCache introduces only shallow-layer probing, the overall additional cost incurred by the probing operation is relatively minimal.

Table 7: **Probing cost of different models.**

| Model | $T_{\text{probe}}$ $(sec)$ | $T_{\text{probe}}/T_{\text{all}}$ |
|---|---|---|
| Flux | 0.19 | 4% |
| WAN 2.1-1.3B | 3.98 | 5% |
| HunyuanVideo | 11.47 | 2% |

## H ETHICAL STATEMENT

In this research, we affirm our commitment to upholding ethical standards in research and promoting responsible innovation. As far as we are aware, our study does not entail any data, methodologies, or applications that pose ethical issues. All experiments and analyses were carried out in accordance with established ethical guidelines, thereby ensuring the integrity and transparency of our research.

## I REPRODUCIBILITY STATEMENT

To ensure full reproducibility of our research and to contribute to the broader academic community, we will publicly release the source code of DiCache. At the current stage, a code implementation of DiCache (for HunyuanVideo and Flux) is available in the zip archive included in the supplementary material. We hope these resources will provide a reference for future caching-based acceleration studies, thereby fostering innovation and accelerating progress within the community.

## J DECLARATION ON LLM USAGE

In this paper, we use LLMs only for minor language polishing.

## K SOURCE CODE AND DEMO VIDEO

We provide the code implementation (for HunyuanVideo and Flux) and a demo video of DiCache for reference in the zip archive included in the supplementary material.

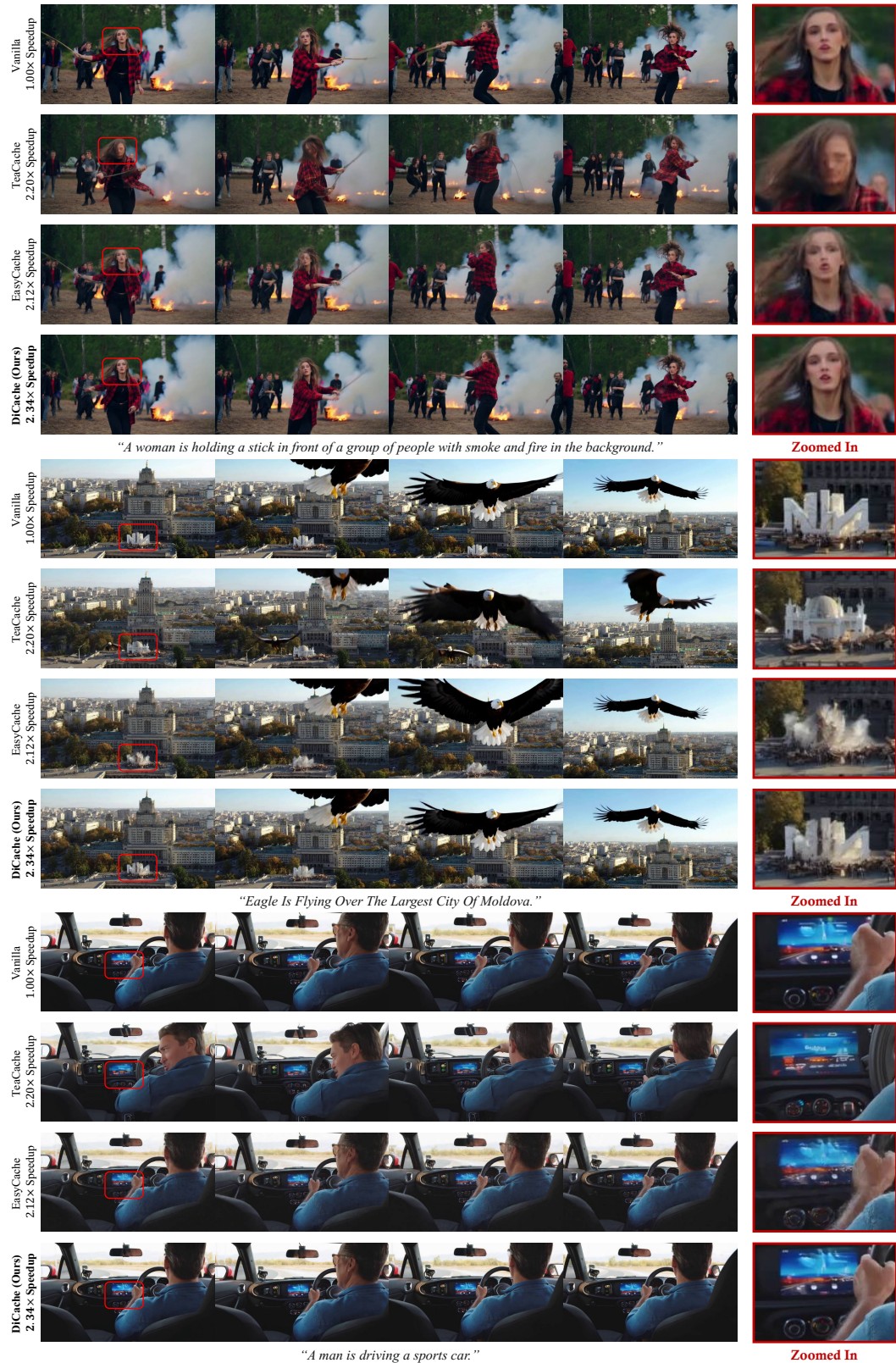

Figure 7: **More qualitative comparison results on HunyuanVideo (1/2). Best viewed zoomed in.**

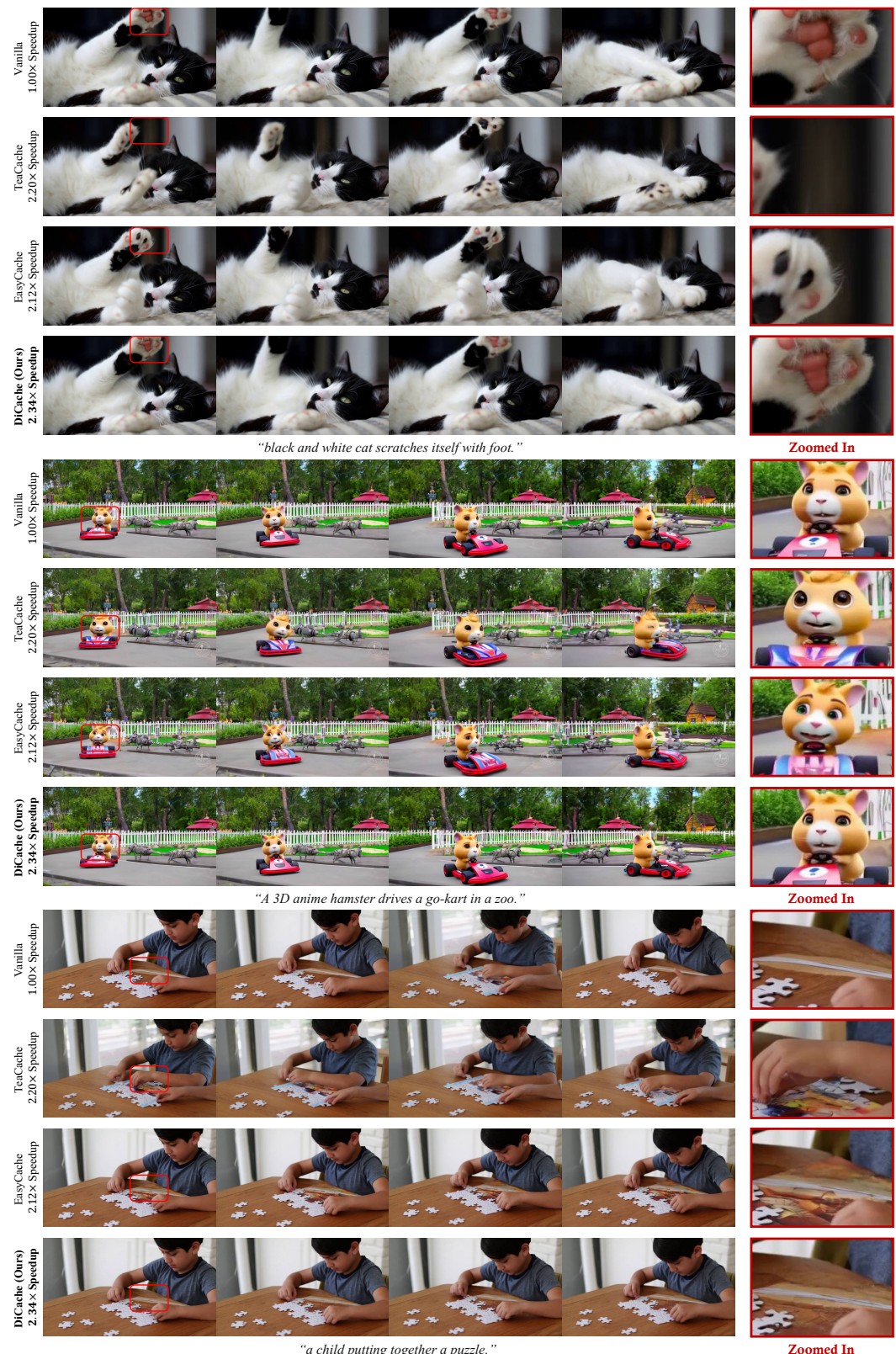

Figure 8: **More qualitative comparison results on HunyuanVideo (2/2). Best viewed zoomed in.**

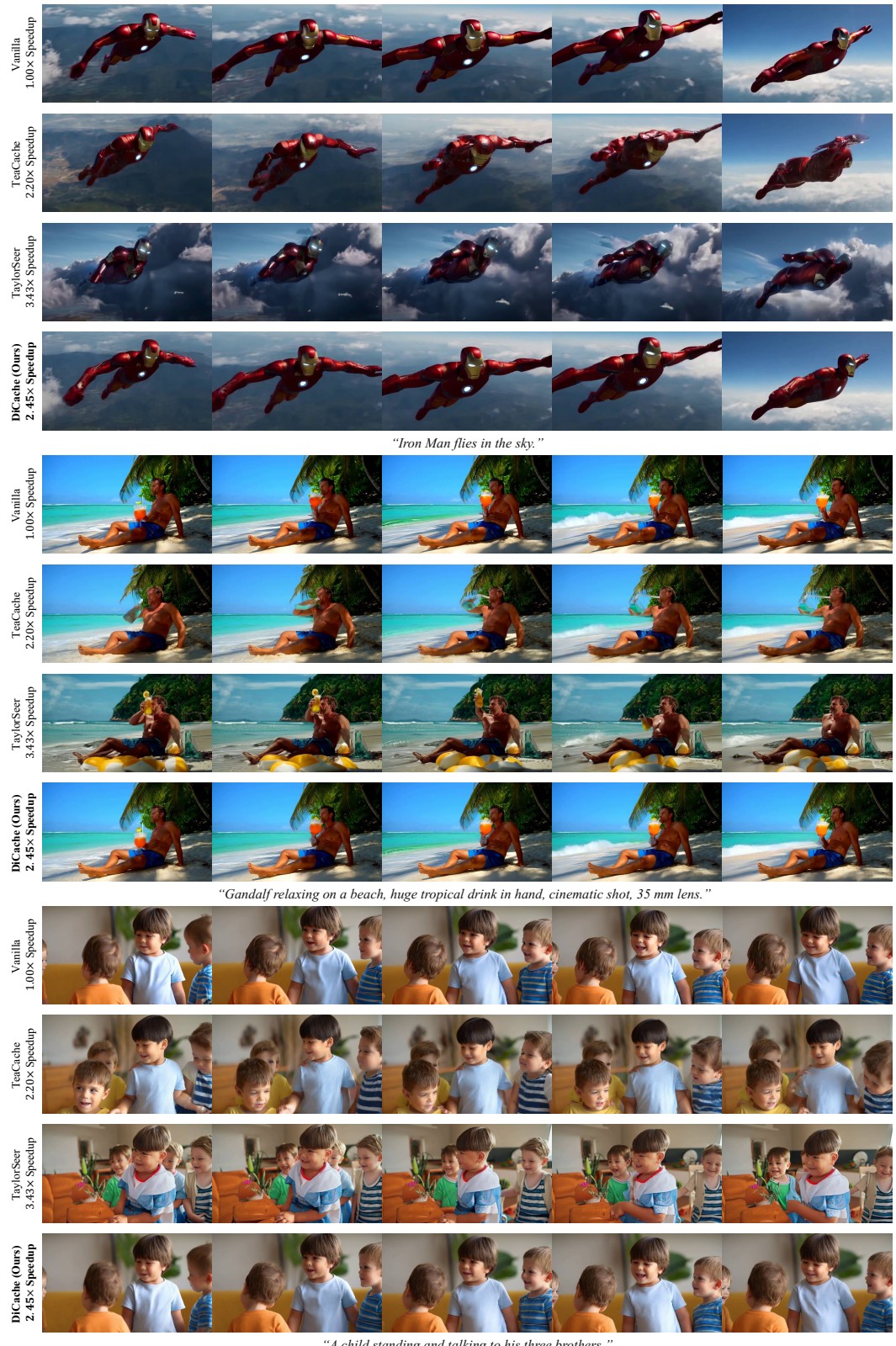

Figure 9: **More qualitative comparison results on WAN 2.1 1.3B. Best viewed zoomed in.**

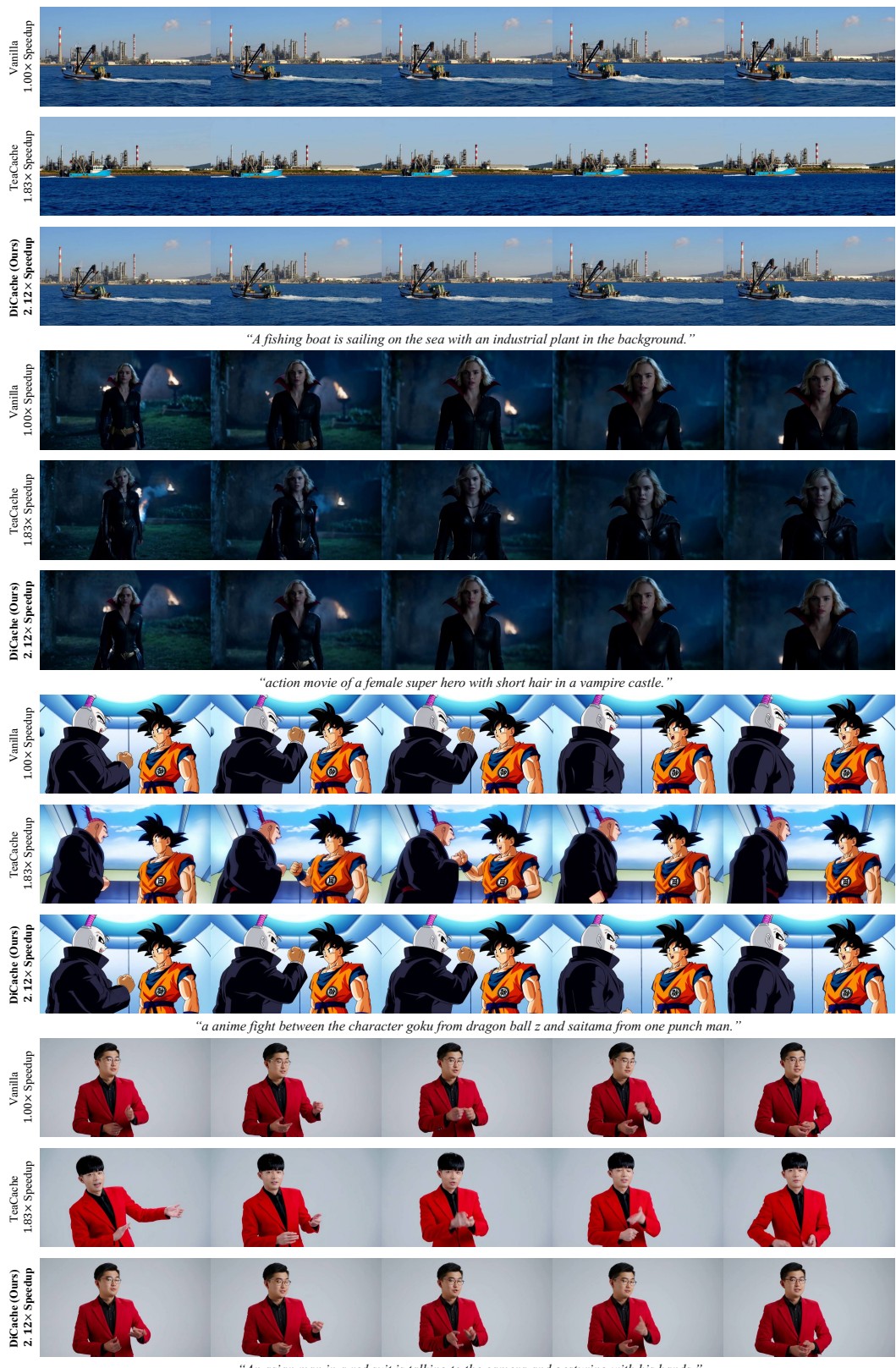

Figure 10: **More qualitative comparison results on WAN 2.1 14B. Best viewed zoomed in.**

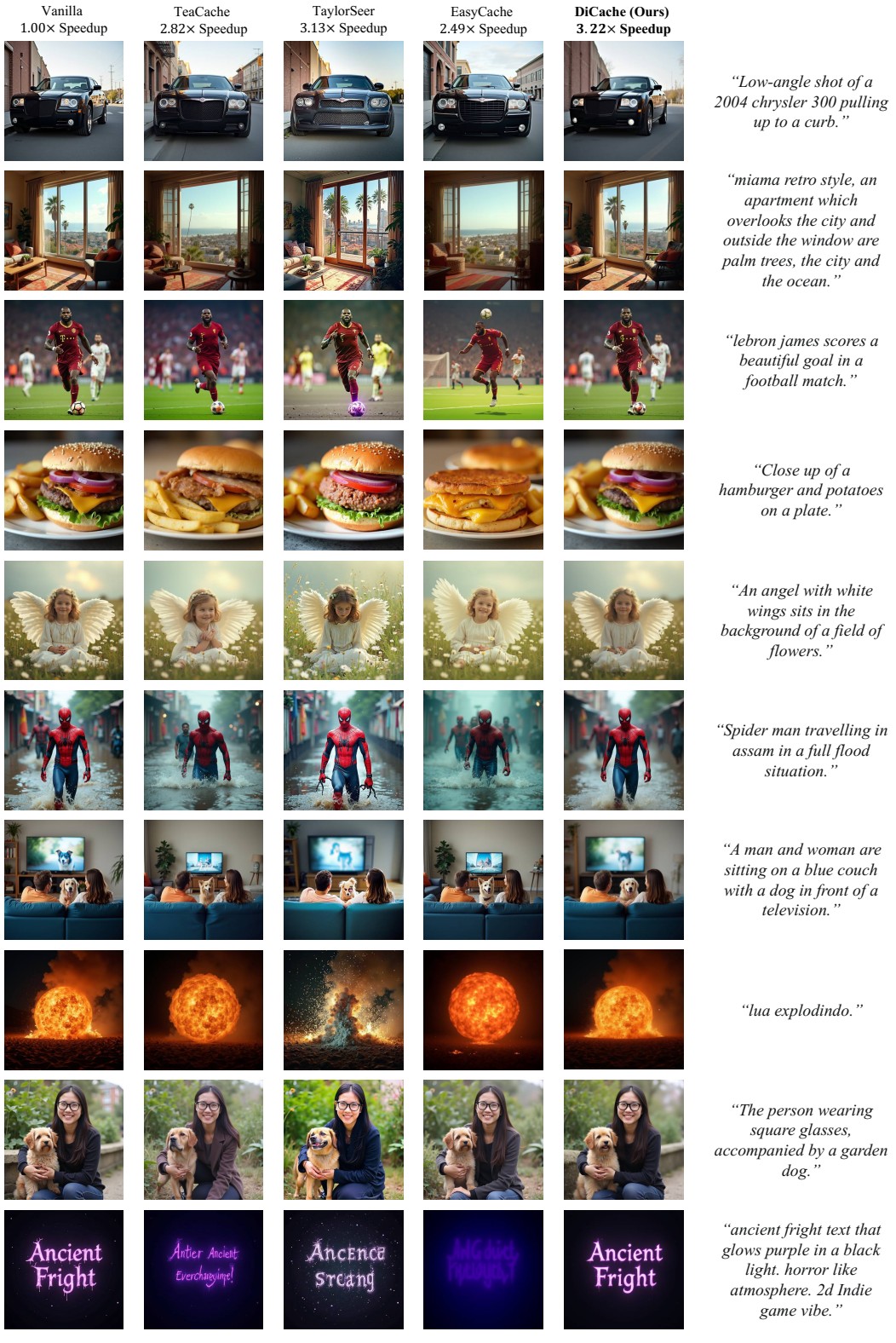

Figure 11: **More qualitative comparison results on Flux (1/2). Best viewed zoomed in.**

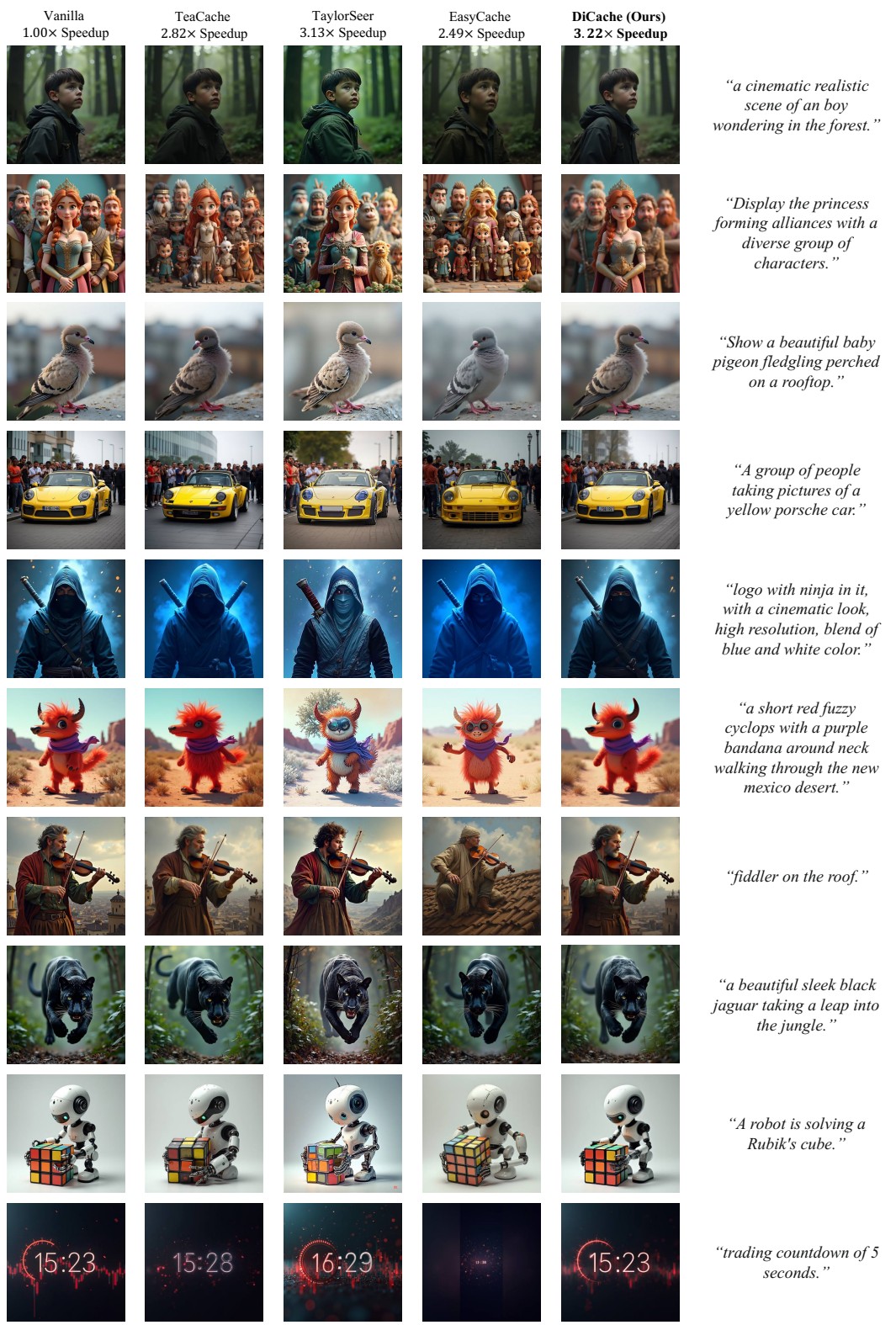

Figure 12: **More qualitative comparison results on Flux (2/2). Best viewed zoomed in.**

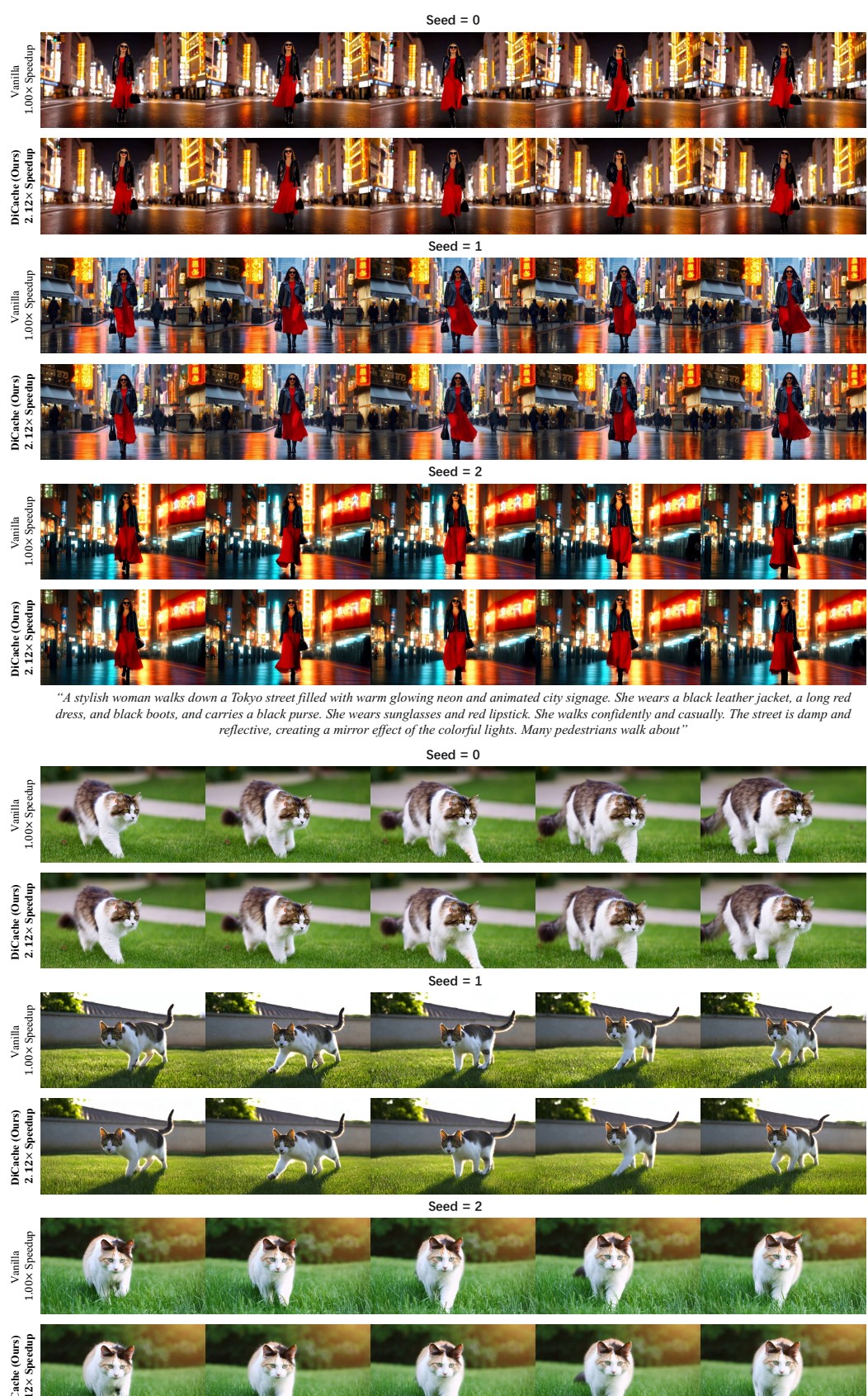

*"A stylish woman walks down a Tokyo street filled with warm glowing neon and animated city signage. She wears a black leather jacket, a long red dress, and black boots, and carries a black purse. She wears sunglasses and red lipstick. She walks confidently and casually. The street is damp and reflective, creating a mirror effect of the colorful lights. Many pedestrians walk about"*

*"A cat walks on the grass, realistic style."*

Figure 13: **Generated results using same prompts and different seeds on WAN 2.1 14B.**

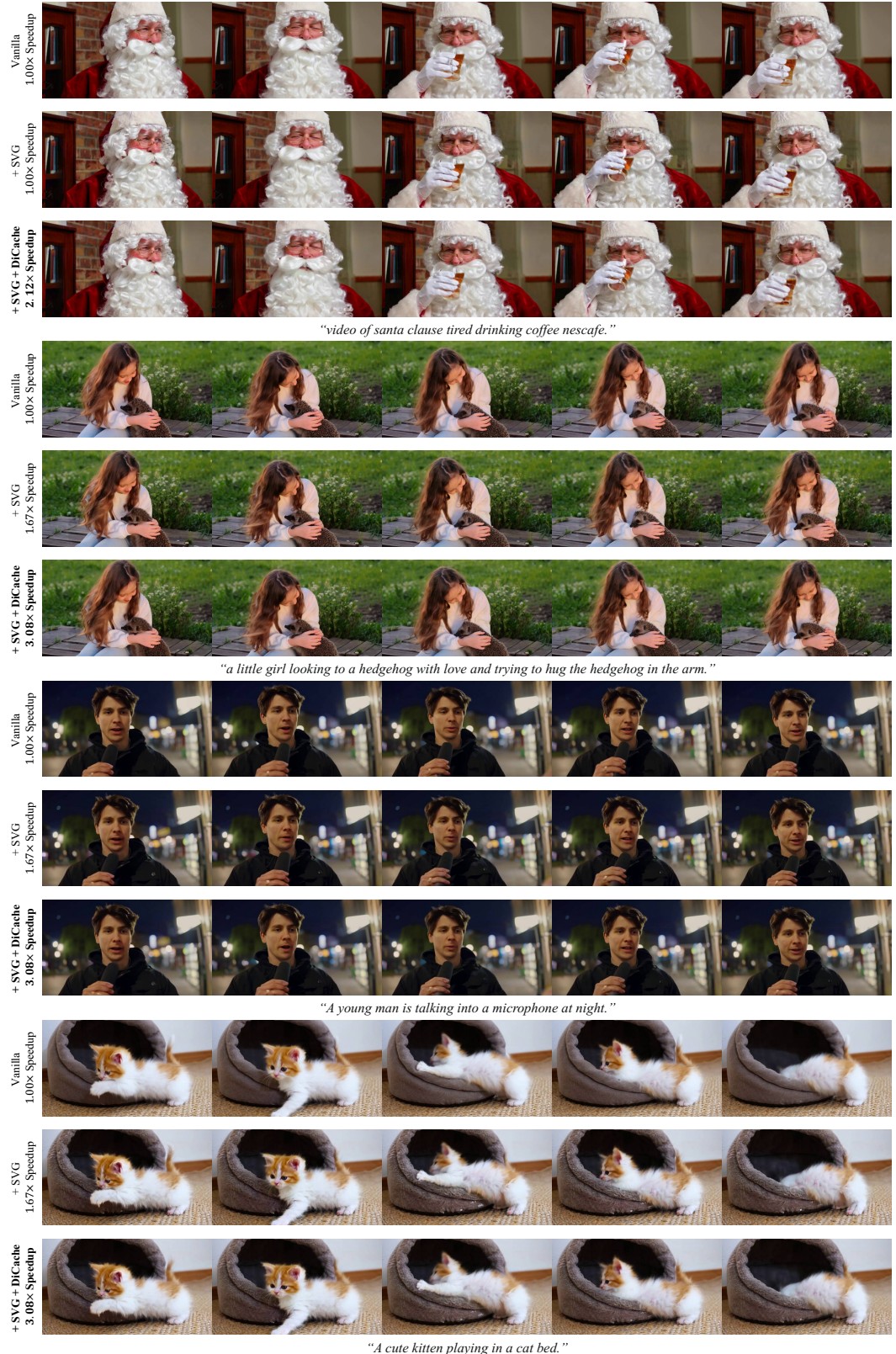

Figure 14: **DiCache can be integrated with the sparse attention method Sparse VideoGen.**

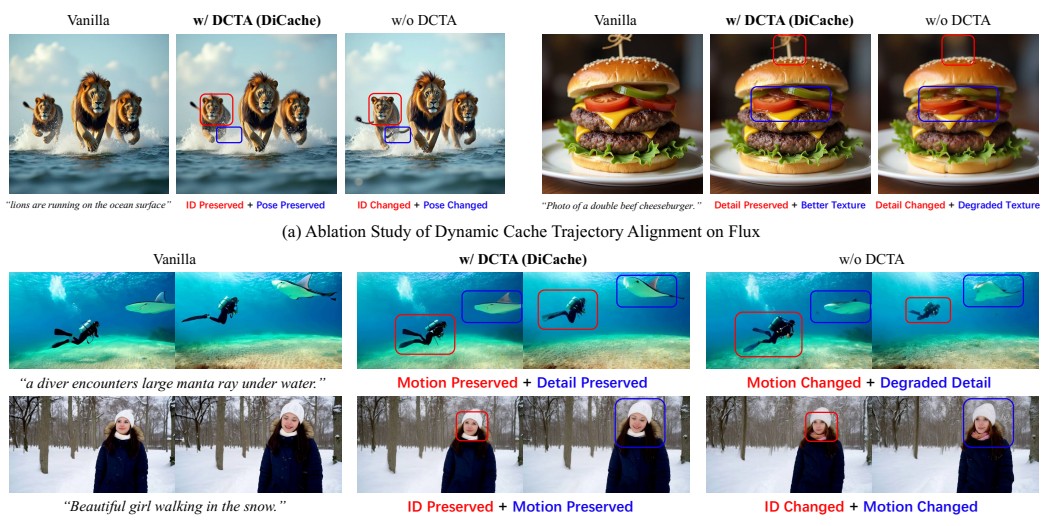

(a) Ablation Study of Dynamic Cache Trajectory Alignment on Flux

(b) Ablation Study of Dynamic Cache Trajectory Alignment on WAN 2.1

Figure 15: **Additional ablation experiments on Dynamic Cache Trajectory Alignment (DCTA).**

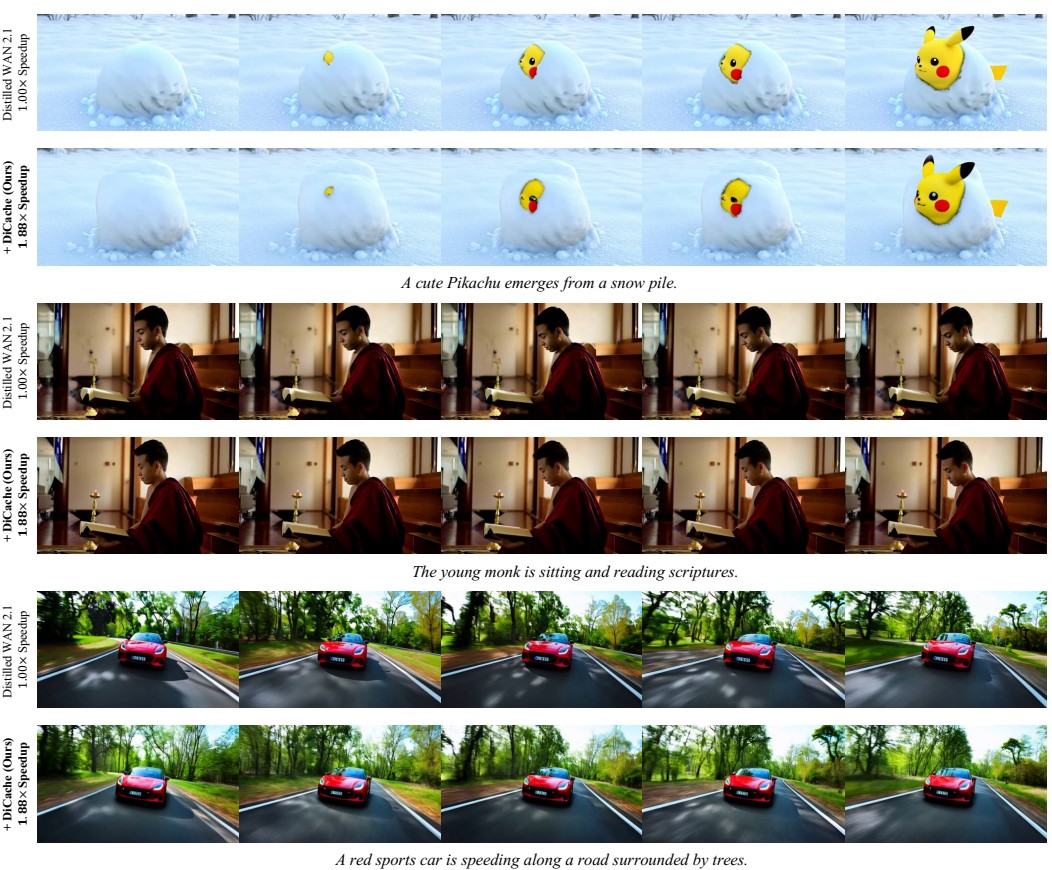

Figure 16: **Performance of DiCache on the distilled WAN 2.1 14B model.**

Table 8: The text prompts for each figure are listed sequentially, following the order from left to right and top to bottom.

| Figure | Text Prompt |
|---|---|
| Figure 1 | A teddy bear is playing drum kit in Times Square.
Harry Potter is talking to his classmate.
Christmas village, cute, snowing, nighttime, cinematic illumination, starry night
a long eared cat with stripes walking and blinking towards the camera on a hill |
| Figure 5 | cowboy chopping a tree with an axe
a beautiful girl in the garden.
Sunset view of a city with mountains in the background. |
| Figure 6 | A dog walking through the streets of New Orleans.
a video game maintenance technician arranges a game on the counter.
racoon eating mango on the beach at the sunset.
a young girl practicing yoga under a large tree. |
| Figure 7 | A woman is holding a stick in front of a group of people with smoke and fire in the background.
Eagle Is Flying Over The Largest City Of Moldova.
A man is driving a sports car. |
| Figure 8 | black and white cat scratches itself with foot.
A 3D anime hamster drives a go-kart in a zoo.
a child putting together a puzzle. |
| Figure 9 | Iron Man flies in the sky.
Gandalf relaxing on a beach, huge tropical drink in hand, cinematic shot, 35 mm lens.
A child standing and talking to his three brothers. |
| Figure 10 | A fishing boat is sailing on the sea with an industrial plant in the background.
action movie of a female super hero with short hair in a vampire castle.
a anime fight between the character goku from dragon ball z and saitama from one punch man.
An asian man in a red suit is talking to the camera and gesturing with his hands. |
| Figure 11 | Low-angle shot of a 2004 chrysler 300 pulling up to a curb.
miama retro style, an apartment which overlooks the city and outside the window are palm trees, the city and the ocean.
lebron james scores a beautiful goal in a football match.
Close up of a hamburger and potatoes on a plate.
An angel with white wings sits in the background of a field of flowers.
Spider man travelling in assam in a full flood situation.
A man and woman are sitting on a blue couch with a dog in front of a television.
lua explodindo.
The person wearing square glasses, accompanied by a garden dog.
ancient fright text that glows purple in a black light. horror like atmosphere. 2d Indie game vibe. |
| Figure 12 | a cinematic realistic scene of an boy wondering in the forest.
Display the princess forming alliances with a diverse group of characters.
Show a beautiful baby pigeon fledgling perched on a rooftop.
A group of people taking pictures of a yellow porsche car.
logo with ninja in it, with a cinematic look, high resolution, blend of blue and white color.
a short red fuzzy cyclops with a purple bandana around neck walking through the new mexico desert.
fiddler on the roof.
a beautiful sleek black jaguar taking a leap into the jungle.
A robot is solving a Rubik's cube.
trading countdown of 5 seconds. |
| Figure 13 | A stylish woman walks down a Tokyo street filled with warm glowing neon and animated city signage. She wears a black leather jacket, a long red dress, and black boots, and carries a black purse. She wears sunglasses and red lipstick. She walks confidently and casually. The street is damp and reflective, creating a mirror effect of the colorful lights. Many pedestrians walk about
A cat walks on the grass, realistic style. |
| Figure 14 | video of santa clause tired drinking coffee nescafe.
a little girl looking to a hedgehog with love and trying to hug the hedgehog in the arm.
A young man is talking into a microphone at night.
A cute kitten playing in a cat bed. |
| Figure 15 | lions are running on the ocean surface
Photo of a double beef cheeseburger.
a diver encounters large manta ray under water.
Beautiful girl walking in the snow. |
| Figure 16 | A cute Pikachu emerges from a snow pile.
The young monk is sitting and reading scriptures.
A red sports car is speeding along a road surrounded by trees. |

