# OpenReview forum: "DiCache: Let Diffusion Model Determine Its Own Cache"
_ICLR.cc/2026/Conference — ICLR 2026 Poster_

### Official Review · Reviewer_1sUf · 2025-10-25

**Soundness:** 2
**Presentation:** 2
**Contribution:** 3
**Rating:** 4
**Confidence:** 4

**Summary:**

This paper proposes a caching mechanism for Diffusion Transformers (DiT). The core idea is to use shallow-layer residual probing to estimate deep-layer feature caching. The author verified DiCache on both text-to-image and text-to-video tasks and obtained superior results compared with the baseline method. In summary, I would say this work introduces a simple yet effective DiT caching method that is easy to use for the community. However, more experimental analysis is expected to make the work more solid and convincing.

**Strengths:**

1. The motivation of this paper is clear. Empirical findings are quite convincing and interesting.
2. This method is easy to use and can be integrated with other intra-step diffusion acceleration methods (such as the sparse-attention-based method).
3. On one hand, the speedup is non-trivial which is especially useful for video diffusion model acceleration. On the other hand, the quality loss is marginal and it can maintain high consistency with the vanilla samples.

**Weaknesses:**

The main weakness of this paper lies in the presentation and insufficient experimental analysis. Please see the questions below.

Some advice on the presentation:
1. I strongly recommend redesigning Figure 2 as it increases the understanding cost.
2. Axes numbers in Figure 3 are too small.
3. Abbreviations are better introduced at the first time they appear, such as DCTA.

**Questions:**

1. Since Wan2.1 and HunyuanVideo are tested, I suggest that the authors provide VBench results on the t2v task. Evaluating generated video quality using image-oriented metrics is not convincing enough.
2. Compared with baseline methods such as TaylorSeer, DiCache uses fewer text-to-image metrics, making the evaluation not comprehensive enough. For example, how does DiCache affect the text-image alignment?
3. Since m is set to 1 for all experiments and the author claimed $m\in [1,2,3] $ is sufficient, I suggest that the author show corresponding diagrams in Figures 3 and 4, instead of only showing m=5.
4.  What is the recomputation rate under different parameter settings? In other words, is $\delta$ hard to tune for different DiT models? It seems this value differs across the DiT models tested.
5. For DCTA, does it mean the current residual is continuously computed from the two most recent recomputed timesteps? How much additional overhead (in terms of FLOPS and latency) will this cause?
6. How does DiCache perform on distilled models?
7. What's the source of text-to-video prompts?

I will raise my score when concerns are mostly addressed.

---

> ### Author Response · Authors · 2025-11-19
> **Response to Reviewer 1sUf (1/3)**
>
> We are greatly encouraged to see that you found our motivation clear, empirical findings convincing and interesting, method both effective and easy to use, and achieving significant speedup with minimal quality loss. We sincerely thank you for your valuable suggestions, which certainly help improve our work. We have accordingly refined our paper as follows:
>
> ### **Q1: Advice on the presentation.**
> **A1:** Thank you for your kind suggestions, and we sincerely apologize for any misunderstanding that may have been caused. We have (i) thoroughly redesigned Figure 2 for improved clarity (as shown in Figure 18 in the appendix, and we will update the main paper accordingly in the final version); (ii) enlarged the axes in Figure 3 for better presentation; (iii) introduced the abbreviation for Dynamic Cache Trajectory Alignment (DCTA) at its first occurrence; and (iv) fixed other minor typos in the paper. Please refer to the revised manuscript for details.
>
> ### **Q2: Provide VBench results on the T2V task.**
> **A2:** Thank you for your valuable suggestion. We have supplemented the comparison results for WAN 2.1-1.3B and HunyuanVideo on VBench [1], using its standard evaluation prompt set. The results are presented in the tables below. Please note that, even though TaylorSeer achieves the highest VBench Score on WAN 2.1, its generated outputs usually deviate significantly from the original content, as evidenced by our quantitative results in the main paper (we also include them in the tables below) and the qualitative comparison depicted in Figure 6 in the main paper and Figure 9/11/12 in the appendix.
>
> **Table1: Quantitative Results for WAN 2.1**
>
> | Method | LPIPS↓   | SSIM↑    | PSNR↑    | VBench Score↑ |
> |:---------------|:--------:|:-------:|:-------:|:------------:|
> | Vanilla        | --      | --      | --      | 82.44        |
> | TeaCache       | 0.2161  | 0.8226  | 20.97   | 81.04        |
> | TaylorSeer     | 0.5214  | 0.5485  | 14.32   | **81.72**    |
> | EasyCache      | _0.2013_| _0.8562_| _24.80_ | 80.96        |
> | **DiCache (Ours)** | **0.1734** | **0.8885** | **26.45** | _81.46_      |
>
> **Table2: Quantitative Results for HunyuanVideo**
>
> | Method         | LPIPS↓   | SSIM↑    | PSNR↑    | VBench Score↑ |
> |:---------------|:--------:|:-------:|:-------:|:------------:|
> | Vanilla        | --      | --      | --      | 82.09        |
> | TeaCache       | 0.2898  | 0.8015  | 22.01   | 81.20        |
> | TaylorSeer     | OOM     | OOM     | OOM     | OOM          |
> | EasyCache      | _0.1558_| _0.9270_| _30.71_ | _81.47_      |
> | **DiCache (Ours)** | **0.1492** | **0.9396** | **32.79** | **81.85**    |
>
> ### **Q3: Add more text-to-image metrics to make a comprehensive evaluation.**
> **A3:** Thanks for your valuable suggestion. We have incorporated CLIP Score [2] and ImageReward [3] as supplementary text-to-image evaluation metrics to assess text–image alignment and visual quality. Following TaylorSeer, we resampled 200 high-quality text prompts from DrawBench [4] and conducted experiments using Flux.1.0-dev. The methods in the table below are evaluated on the DrawBench prompts, and all metrics are recomputed accordingly. It is worth noting that although TaylorSeer achieves relatively good performance on CLIP Score and ImageReward, its generated images often exhibit show low similarity to the original outputs (sometimes even with abnormal color contrast), as reflected in the LPIPS, SSIM, and PSNR metrics in the table below, as well as the qualitative comparisons in Figure 6/11/12.
>
> **Table3: Quantitative Results for Flux on DrawBench**
>
> | Method        |   LPIPS↓   |   SSIM↑   |   PSNR↑  | CLIP Score↑ | ImageReward↑ |
> | :------------ | :-------: | :------: | :-----: | :--------: | :---------: |
> | Vanilla       |     -     |    -     |    -    |   18.85    |   1.0804    |
> | TeaCache      |  0.3886   |  0.7148  |  18.12  |   18.52    |   0.9785    |
> | TaylorSeer    |  0.4554   |  0.6680  |  15.81  | **19.08**  | *1.0154*    |
> | EasyCache     |  0.3105   |  0.7677  |  19.43  |   18.22    |   0.8995    |
> | **DiCache (Ours)** | **0.2540** | **0.8176** | **22.65** | *18.71* | **1.0380**  |
>
> ### **Q4: Show diagrams for $m\in[1,3]$ in Figures 3 and 4, instead of only showing $m=5$.**
> **A4:** Thanks for your constructive suggestion. We have supplemented the visualizations of the feature differences for DiT layers 1/3 and the corresponding feature trajectories in Figures 15 and 16 of the appendix, respectively.

---

> ### Author Response · Authors · 2025-11-19
> **Response to Reviewer 1sUf (2/3)**
>
> ### **Q5: What is the recomputation rate under different parameter settings? Is $\delta$ hard to tune for different DiT models?**
> **A5:** Thank you for your valuable comment. We report the recomputation ratio of HunyuanVideo under different reuse thresholds, as shown in the table below.
>
> **Table4: Recomputation Ratio of HunyuanVideo under Different $\delta$**
>
> | Reuse Threshold | Recomputation Ratio |
> | :-------------- | :-----------------: |
> | $\delta=0.20$        |        34 %         |
> | $\delta=0.15$         |        38 %         |
> | $\delta=0.10$         |        42 %         |
> | $\delta=0.08$         |        46 %         |
> | $\delta=0.05$         |        56 %         |
>
> Admittedly, the recommended value of the reuse threshold $\delta$ varies across models (these variations arise from the differences in training data and architectural design among different DiT models), but the differences are relatively minor. A $\delta \in [0.1, 0.5]$ is recommended since it has shown effectiveness across most tested models. Therefore, **tuning $\delta$ is not difficult in practice**.  Furthermore, once the value of $\delta$ is determined, it exhibits strong stability across different samples within the same model. Specifically, we sample 1K diverse prompts (the same evluation set as the main paper) from the LAION-5B dataset and use Flux for image generation. The results demonstrate that DiCache not only maintains the lowest LPIPS but also exhibits the lowest LPIPS variance compared to state-of-the-art methods (as shown in the table below), indicating its stable performance across diverse samples and strong generalization capability.
>
> **Table5: Mean and Variance of LPIPS on Flux**
>
> | Method       | Mean(LPIPS)↓ | Variance(LPIPS)↓ |
> |:---------------|:--------------:|:------------------:|
> | TeaCache     | 0.4053       | 0.010            |
> | TaylorSeer   | 0.4709       | 0.007            |
> | EasyCache    | 0.3049       | 0.008            |
> | **DiCache (Ours)** | **0.2704** | **0.004**       |
>
> In addition, since **acceleration for diffusion models is a topic highly dependent on practical needs**, some users may prioritize higher speedups, while others may prefer better visual fidelity. Therefore, adjusting $\delta$ at inference time according to practical needs provides greater flexibility and better accommodates diverse user requirements.
>
> ### **Q6: For DCTA, does it mean the current residual is continuously computed from the two most recent recomputed timesteps? How much additional overhead (in terms of FLOPS and latency) will this cause?**
> **A6:** Thanks for your constructive comment. The purpose of Dynamic Cache Trajectory Alignment (DCTA) is to leverage multi-step historical caches to dynamically estimate the feature at the current timestep, under the guidance of the shallow-layer feature trajectory. In our implementation, the two most recently computed caches are utilized to adaptively extrapolate the current residual:
>
> $$r_t = r_{t_\beta} + \gamma_t(r_{t_\alpha}-r_{t_\beta}),$$
>
> where the trajectory parameter $\gamma_t$ is dynamically solved using shallow-layer features (see Equation 10 and 11 in the main paper). **Since this is essentially a simple feature addition–subtraction operation without any projection or other complex computations, it introduces minimal extra computational overhead**. As shown in our ablation study (also included in the table below), we analyze both the visual quality metrics and inference latency differences of HunyuanVideo with and without DCTA. It can be observed that DCTA introduces minimal additional latency, while consistently providing performance gains.
>
> **Table6: Ablation Results of DCTA on HunyuanVideo**
>
> | Method   |  LPIPS↓  |  SSIM↑   |  PSNR↑  | Speedup↑ | Latency (sec)↓ |
> | :------- | :-----: | :-----: | :----: | :-----: | :-----------: |
> | DiCache w/o DCTA | 0.1517  | 0.9314  | 31.98  | **2.34×** |  **506.82**   |
> | **DiCache w/ DCTA** | **0.1492** | **0.9396** | **32.79** | **2.34×** | 507.24 |

---

> ### Author Response · Authors · 2025-11-19
> **Response to Reviewer 1sUf (3/3)**
>
> ### **Q7: How does DiCache perform on distilled models?**
> **A7:** Thank you for your insightful suggestion. Here, we combine DiCache with the distilled version of WAN 2.1-14B (by AccVideo [5], with 10 sampling steps), achieving a notable acceleration. The quantitative evaluation is presented in the table below, while the qualitative results are shown in Figure 19 in the revised manuscript. We use 300 prompts collected from the VideoDPO [6] and VideoFeedback [7] datasets (refined by GPT-4o [8] for better quality) as the evaluation benchmark. Experiments in this section are conducted on a single NVIDIA H200 GPU. The reuse threhold $\delta$ and the probe depth $m$ are set to (0.2, 1).
>
> **Table7: Performance of DiCache on the Distilled WAN 2.1-14B**
>
> | Method           |  LPIPS↓  |  SSIM↑   |  PSNR↑  | Speedup↑ | Latency (sec)↓ |
> | :--------------- | :-----: | :-----: | :----: | :-----: | :-----------: |
> | Vanilla AccVideo |    –    |    –    |   –    | 1.00×   |    48.24      |
> | **+ DiCache**    | 0.2737  | 0.7432  | 19.72  | 1.56×   |    30.98      |
>
> Even though DiCache demonstrates some effectiveness on the distilled WAN 2.1, we must acknowledge that, due to the reduced number of inference steps in the distilled model, the flow velocity differences between consecutive timesteps are relatively large, which can limit the acceleration achievable by caching-based methods, as they typically rely on the similarity of flow velocities across adjacent timesteps to skip computations.
>
> ### **Q8: What's the source of text-to-video prompts?**
> **A8:** Thank you for your valuable comment. The text-to-video prompts used in our main experiments are collected from the VideoDPO [6] and VideoFeedback [7] datasets. For higher prompt quality, we further polish them using GPT-4o [8]. We have added a description of our T2V evaluation dataset in the revised manuscript.
>
>
> ### **Reference**
>
> [1] Huang Z, He Y, Yu J, et al. Vbench: Comprehensive benchmark suite for video generative models[C]//Proceedings of the IEEE/CVF Conference on Computer Vision and Pattern Recognition. 2024: 21807-21818.
>
> [2] Radford A, Kim J W, Hallacy C, et al. Learning transferable visual models from natural language supervision[C]//International conference on machine learning. PmLR, 2021: 8748-8763.
>
> [3] Xu J, Liu X, Wu Y, et al. Imagereward: Learning and evaluating human preferences for text-to-image generation[J]. Advances in Neural Information Processing Systems, 2023, 36: 15903-15935.
>
> [4] Saharia C, Chan W, Saxena S, et al. Photorealistic text-to-image diffusion models with deep language understanding[J]. Advances in neural information processing systems, 2022, 35: 36479-36494.
>
> [5] Zhang H, Chen X, Wang Y, et al. Accvideo: Accelerating video diffusion model with synthetic dataset[J]. arXiv preprint arXiv:2503.19462, 2025.
>
> [6] Liu R, Wu H, Zheng Z, et al. Videodpo: Omni-preference alignment for video diffusion generation[C]//Proceedings of the Computer Vision and Pattern Recognition Conference. 2025: 8009-8019.
>
> [7] He X, Jiang D, Zhang G, et al. Videoscore: Building automatic metrics to simulate fine-grained human feedback for video generation[C]//Proceedings of the 2024 Conference on Empirical Methods in Natural Language Processing. 2024: 2105-2123.
>
> [8] Hurst A, Lerer A, Goucher A P, et al. Gpt-4o system card[J]. arXiv preprint arXiv:2410.21276, 2024.

---

> ### Comment · Reviewer_1sUf · 2025-11-24
>
> Thanks authors for the detailed response and additional experimental results. Most of my concerns have been addressed. I'll raise my score.

---

> > ### Author Response · Authors · 2025-11-24
> >
> > Dear Reviewer 1sUf,
> >
> > Thank you for your positive and encouraging feedback on our work!
> >
> > We are glad to hear that our rebuttal has addressed your concerns. The corresponding revisions will be incorporated into the final version of the paper.
> >
> > We sincerely appreciate your time and effort in providing such meticulous reviews and insightful comments.
> >
> > Best regards,
> >
> > Authors

---

### Official Review · Reviewer_MEuZ · 2025-10-26

**Soundness:** 2
**Presentation:** 2
**Contribution:** 3
**Rating:** 6
**Confidence:** 4

**Summary:**

This paper proposes DiCache, a unified probe-driven framework that adaptively schedules and utilizes cache during diffusion inference, achieving efficient acceleration without additional training.

**Strengths:**

1. The paper addresses the two fundamental challenges of cache-based acceleration through a unified probe-driven framework, which reduces reliance on empirical heuristics and offline calibration.

2. DiCache can be further combined with Sparse VideoGen to achieve additional acceleration, demonstrating its complementarity with sparse attention techniques.

3. The authors empirically observe a strong correlation between shallow-layer feature differences and deep-layer residuals, and find that features across different DiT blocks exhibit similar trajectories during the sampling process, providing the foundation for the probe-based error estimation and trajectory-based cache blending.

**Weaknesses:**

1. The coverage of baselines is somewhat limited. Although the experimental tables include TeaCache, EasyCache, TaylorSeer, and ToCa, the Related Work section also discusses other comparable methods such as FasterCache, FORA, and Δ-DiT. Incorporating these methods into the quantitative comparison tables would make the empirical positioning of the proposed approach more complete.

2. The paper primarily evaluates performance using similarity metrics (LPIPS, SSIM, and PSNR) with respect to the outputs of the original model, along with the inference speedup ratio. However, it provides limited analysis of broader perceptual quality or downstream task metrics. Since DiCache does not demonstrate particularly strong fidelity compared to the base model, including perceptual quality evaluations such as VBench scores and user studies would be necessary to provide a more comprehensive assessment.

**Questions:**

See Weakness

---

> ### Author Response · Authors · 2025-11-19
> **Response to Reviewer MEuZ (1/2)**
>
> We are greatly encouraged to see that you found our method addressing the two fundamental challenges of cache-based acceleration, compatible with other acceleration techniques like sparse attention, and is well-grounded in feature observations. We sincerely thank you for your valuable suggestions, which certainly help improve our work. We have accordingly refined our paper as follows:
>
> ### **Q1: The Related Work section also discusses other comparable methods such as FasterCache, FORA, and $\Delta$-DiT. Incorporating these methods into the quantitative comparison tables would make the empirical positioning of the proposed approach more complete.**
> **A1:** Thank you for your constructive suggestion. Since the official implementations of FasterCache, FORA, and $\Delta$-DiT were developed on earlier DiT models (e.g, OpenSora [1] and DiT for ImageNet [2]), we re-implemented all three methods on the leading DiT model Flux to ensure a fair comparison. As modern models such as Flux and HunyuanVideo often adopt classifier-free guidance (CFG) distillation, the CFG-cache of FasterCache is disabled. Experiments in this section are conducted on an NVIDIA A800 80GB GPU. As can be observed in the table below, our DiCache significantly outperforms FasterCache, FORA, and $\Delta$-DiT in both efficiency and generation quality.
>
> **Table1: Quantitative Comparison with More Baselines**
>
> | Method         | LPIPS↓   | SSIM↑    | PSNR↑    | Speedup↑   | Latency (sec)↓ |
> |----------------|:---:|:---:|:---:|:---:|:---:|
> | Vanilla        | -       | -       | -       | 1.00×     | 15.11        |
> | $\Delta$-DiT          | 0.5535  | 0.6058  | 15.57   | 1.84×      | 8.21         |
> | FORA           | 0.4716  | 0.6396  | 17.04   | _2.35×_   | _6.43_       |
> | FasterCache    | _0.3478_| _0.7490_| _20.22_ | 1.52×      | 9.94         |
> | **DiCache (Ours)** | **0.2704** | **0.8211** | **22.39** | **3.22×**   | **4.69**     |
>
> ### **Q2: Including perceptual quality evaluations such as VBench scores and user studies would be necessary to provide a more comprehensive assessment.**
> **A2:** Thank you for your valuable suggestion. We have supplemented the comparison results for WAN 2.1-1.3B and HunyuanVideo on VBench [3], using its standard evaluation prompt set. The results are presented in the tables below. Please note that, even though TaylorSeer achieves the highest VBench Score on WAN 2.1, its generated outputs usually deviate significantly from the original content, as evidenced by our quantitative results in the main paper (we also included them in the tables below) and the qualitative comparison depicted in Figure 6 in the main paper and Figure 9 in the appendix.
>
> **Table2: Quantitative Results for WAN 2.1**
>
> | Method | LPIPS↓   | SSIM↑    | PSNR↑    | VBench Score↑ |
> |:---------------|:--------:|:-------:|:-------:|:------------:|
> | Vanilla        | --      | --      | --      | 82.44        |
> | TeaCache       | 0.2161  | 0.8226  | 20.97   | 81.04        |
> | TaylorSeer     | 0.5214  | 0.5485  | 14.32   | **81.72**    |
> | EasyCache      | _0.2013_| _0.8562_| _24.80_ | 80.96        |
> | **DiCache (Ours)** | **0.1734** | **0.8885** | **26.45** | _81.46_      |
>
> **Table3: Quantitative Results for HunyuanVideo**
>
> | Method         | LPIPS↓   | SSIM↑    | PSNR↑    | VBench Score↑ |
> |:---------------|:--------:|:-------:|:-------:|:------------:|
> | Vanilla        | --      | --      | --      | 82.09        |
> | TeaCache       | 0.2898  | 0.8015  | 22.01   | 81.20        |
> | TaylorSeer     | OOM     | OOM     | OOM     | OOM          |
> | EasyCache      | _0.1558_| _0.9270_| _30.71_ | _81.47_      |
> | **DiCache (Ours)** | **0.1492** | **0.9396** | **32.79** | **81.85**    |
>
> Moreover, we have conducted a user study with 30 participants to assess _Video Quality_ and _Visual Similarity_ (regarding the original results) on the T2V model WAN 2.1. Each participant is presented with 10 sets of videos, with each set containing videos generated by different caching methods (TeaCache, TaylorSeer, EasyCache and DiCache) using the same text prompt and random seed. For each video set, participants are required to select the video they perceive as superior based on overall _Video Quality_ and _Visual Similarity_ (with respect to a given original video), respectively. The videos were presented in a randomized order to reduce potential bias, and participants were allowed ample time to review each set before making their selections. As shown in the tables below, our proposed DiCache receives the majority of votes in both assessed metrics, demonstrating its superiority over existing state-of-the-art approaches.
>
> **Table4: User Study results for WAN 2.1**
>
> | Method   | Video Quality↑ | Visual Similarity↑ |
> |:---------------|:--------------:|:------------------:|
> | TeaCache   | 11.00%        | 7.33%      |
> | TaylorSeer     | 18.00%        | 2.67%        |
> | EasyCache      | 25.67%        | 29.33%        |
> | **DiCache (Ours)** | **45.33%** | **60.67%**        |

---

> ### Author Response · Authors · 2025-11-19
> **Response to Reviewer MEuZ (2/2)**
>
> ### **Reference**
>
> [1] Zheng Z, Peng X, Yang T, et al. Open-sora: Democratizing efficient video production for all[J]. arXiv preprint arXiv:2412.20404, 2024.
>
> [2] Peebles W, Xie S. Scalable diffusion models with transformers[C]//Proceedings of the IEEE/CVF international conference on computer vision. 2023: 4195-4205.
>
> [3] Huang Z, He Y, Yu J, et al. Vbench: Comprehensive benchmark suite for video generative models[C]//Proceedings of the IEEE/CVF Conference on Computer Vision and Pattern Recognition. 2024: 21807-21818.

---

> ### Author Response · Authors · 2025-11-27
>
> Dear Reviewer MEuZ,
>
> We sincerely appreciate your time and effort in providing such meticulous reviews and insightful comments.
>
> Could you please kindly let us know if our rebuttal has addressed your concerns?
>
> If you have any further questions regarding our work, we would be delighted to address them.
>
> Best regards,
>
> Authors

---

### Official Review · Reviewer_HkJV · 2025-10-28

**Soundness:** 3
**Presentation:** 4
**Contribution:** 3
**Rating:** 6
**Confidence:** 3

**Summary:**

This work presents DiCache, a training-free, runtime-adaptive caching framework for diffusion models.
In DiCache, the Online Probe Profiling Scheme runs the first few layers to extract shallow probe features and estimate a per-sample caching error, which serves as the threshold for cache reuse.
Dynamic Cache Trajectory Alignment then uses the probe’s estimated progress to select and align deep features from nearby cached steps, reconstructing the current step without recomputing the heavy layers.
As a result, DiCache achieves high speedups with stronger fidelity across WAN 2.1, HunyuanVideo, and Flux.

**Strengths:**

Novel idea on dynamic cache trajectory alignment that effectively and efficiently adapts the cached value to the current layer that reuses it.

training-free and plug-and-play, requiring no model fine-tuning; it works at inference by wrapping around any DiT models.

DiCache consistently achieves faster inference without sacrificing output quality, outperforming prior caching methods on both image and video diffusion models

Clear analysis and ablations with generally smooth writing and informative figures; easy to follow overall.

**Weaknesses:**

1.	**Reliance on Threshold Hyperparameter:** Although DiCache demonstrates effective runtime caching under the reported experimental settings, the chosen probe depth (m) and accumulated caching error threshold (δ) should ideally generalize across different models. Alternatively, the authors could justify that the method’s effectiveness is not sensitive to these hyperparameters to substantiate the “calibration-free” claim. However, the current analysis of both hyperparameters lacks evidence of such generalization. For example, Spearman correlation analyses (e.g., Fig. 3 (d)) across multiple architectures would strengthen this point.
2.	**Hyperparameter Trade-offs:** There exists a strong trade-off among probe depth, reuse threshold, and achieved speedup, as well as between the reuse threshold and output accuracy. This implies that achieving the optimal quality–efficiency balance may require manual hyperparameter tuning, making the approach functionally similar to other calibration-based caching methods rather than being fully self-adaptive.
3.	**Overlap with Prior Works on Adaptive Caching and Probing:** DiCache’s adaptive cache decision, which is based on accumulated probe scores and thresholding, follows the same high-level “accumulate + threshold” mechanism used in TeaCache, and its distance-based error proxy conceptually resembles AdaCache’s feature-change metric. Furthermore, shallow-layer probing and caching is a common technique in transformer models [1, 2]. The authors should clarify the DiT-specific novelty in their cache-layer determination, beyond the incremental combination or adaptation of these existing ideas.
4.	Effectiveness of DCTA: Although DCTA is presented as a major contribution of this work, the breakdown of DiCache’s accuracy in Table 2 raises concern about its actual impact, as the observed improvement appears marginal. It would be helpful to clarify whether the effectiveness of DCTA, while seemingly limited in magnitude, remains consistent across different models.
5.	**Memory Usage Analysis:** The paper lacks quantitative analysis of memory usage, even though higher-order or multi-step caching inevitably trades off between memory consumption and accuracy. Reporting VRAM or feature-map memory profiles would clarify the practical scalability of the proposed method.

[1] LayerSkip: Enabling Early Exit Inference and Self-Speculative Decoding, Mostafa Elhoushi, Akshat Shrivastava, et al., ACL 2024.

[2] Reducing Transformer Key-Value Cache Size with Cross-Layer Attention, William Brandon, Mayank Mishra et al., NeurIPS 2024.

**Questions:**

Mainly listed in the weakness. Below are the additional questions.

1.	**Reasoning on probing:** Explaination on why the first few layers can be so informative for predicting global caching error—e.g., what structural or representational property of DiTs enables such strong shallow–deep feature correlation would strenghten the paper.

2.	**Minor:** (i) Typo: missing space in “…probe feature trajectory,which…” (line 108). (ii) Naming inconsistency between “Online Probe Profiling Scheme” (main text) and “Online Probe Profiling Strategy” (Fig. 2 caption).

---

> ### Author Response · Authors · 2025-11-19
> **Response to Reviewer HkJV (1/2)**
>
> We are greatly encouraged to see that you found our method novel and effective in dynamically aligning cache trajectories, and that you recognized our plug-and-play design, superior quality over
> baselines and clear presentation. We sincerely thank you for your valuable suggestions, which certainly help improve our work. We have accordingly refined our paper as follows:
>
> ### **Q1: Reliance on Threshold Hyperparameter.**
> **A1:** Thanks for your valuable comment. The detailed explanation regarding the two hyperparameters $m$ (probe depth) and $\delta$ (reuse threshold) in DiCache is provided below:
>
> 1. **Probe Depth $m$**. It denotes the number of DiT layers that is probed at each timestep. As shown in Figure 3(d), a very shallow probe depth ($m\in[1,3]$) **across different model architectures (HunyuanVideo and WAN 2.1)** is sufficient to obtain an accurate caching error indicator. **In our main experiments, we use $m=1$ for all three models**. Therefore, the probe depth $m$ serves as a highly generalizable hyperparameter that transfers well across different architectures.
>
> 2. **Reuse Threshold $\delta$**. It represents the tolerance level for caching errors. Admittedly, the recommended value of the reuse threshold $\delta$ varies across models (these variations arise from the differences in training data and architectural design among different DiT models), but the differences are relatively minor. A $\delta \in [0.1, 0.5]$ is recommended since it has shown effectiveness across most tested models. **Furthermore, once the value of $\delta$ is determined, it exhibits strong stability across different samples within the same model.** Specifically, we sample 1K diverse prompts (the same evluation set as the main paper) from the LAION-5B dataset and use Flux for image generation. The results demonstrate that DiCache not only maintains the lowest LPIPS but also exhibits the lowest LPIPS variance compared to state-of-the-art methods (as shown in the table below), indicating its stable performance across diverse samples and strong generalization capability.
>
> **Table1: Mean and Variance of LPIPS on Flux**
>
> | Method       | Mean(LPIPS)↓ | Variance(LPIPS)↓ |
> |:---------------|:--------------:|:------------------:|
> | TeaCache     | 0.4053       | 0.010            |
> | TaylorSeer   | 0.4709       | 0.007            |
> | EasyCache    | 0.3049       | 0.008            |
> | **DiCache (Ours)** | **0.2704** | **0.004**       |
>
> In addition, since **acceleration for diffusion models is a topic highly dependent on practical needs**, some users may prioritize higher speedups, while others may prefer better visual fidelity. Therefore, adjusting $\delta$ at inference time according to practical needs provides greater flexibility and better accommodates diverse user requirements.
>
> ### **Q2: Hyperparameter Trade-offs.**
> **A2:** Thanks for your valuable comment.  As discussed in our response to Question 1, **DiCache does not require extensive manual hyperparameter tuning and is a highly adaptive framework**. For the probe depth $m$, it has been validated that $m \in [1,3]$ is an efficient and highly generalizable choice that transfers well across different model architectures ($m=1$ is adopted for all three models in our main experiments). For the reuse threshold $\delta$, its recommended value generally falls into $[0.1, 0.5]$, which can be easily adjusted in practical applications. Moreover, we would like to emphasize that **the performance–efficiency trade-off of DiCache allows users to flexibly balance acceleration and quality according to their diverse requirements in speeding up diffusion models**. For example, some users may prefer a lower $\delta$ to achieve higher image quality, while others may choose a higher $\delta$ to maximize inference speed. Therefore, these hyperparameters do not have a single “optimal” value and can be flexibly tuned within a reasonable range to meet practical needs.
>
> ### **Q3: Overlap with Prior Works on Adaptive Caching and Probing.**
> **A3:** Thank you for your constructive comment. The "accumulate + threshold" mechanism and "distance-based error proxy" are both intuitive and broadly applicable approaches, and have been widely adopted in several outstanding caching-based works including TeaCache, AdaCache, and EasyCache. While shallow-layer probing has indeed been used in prior researches in transformers, to the best of our knowledge, **we are the first to demonstrate that shallow and deep features for DiT models exhibit similar differences and trajectory patterns on a sample-specific basis, providing a foundation for the application of shallow-layer probing in caching-based diffusion model acceleration**. Building on this novel perspective of probing DiT layers at inference time, we propose the innovative DiCache framework, which significantly outperforms existing caching-based methods in both quality and efficiency.

---

> ### Author Response · Authors · 2025-11-19
> **Response to Reviewer HkJV (2/2)**
>
> ### **Q4: Effectiveness of DCTA.**
> **A4:** Thanks for your valuable suggestion. We have supplemented the ablation studies of Dynamic Cache Trajectory Alignment (DCTA) on both Flux and WAN 2.1 to validate its effectiveness. The quantitative evaluation is presented in the table below, while the qualitative results are shown in Figure 17 in the revised manuscript. It can be observed from these results that DCTA consistently brings stable improvements across all evaluated metrics (LPIPS, SSIM, and PSNR) and delivers notable gain in visual quality across different model architectures.
>
> **Table2: Ablation Result of DCTA on Flux**
>
> | Method       | LPIPS↓   | SSIM↑    | PSNR↑    |
> |--------------|---------|---------|---------|
> | w/o DCTA     | 0.2847  | 0.8053  | 21.87   |
> | **w/ DCTA**  | **0.2704** | **0.8211** | **22.39** |
>
> **Table3: Ablation Result of DCTA on WAN 2.1**
>
> | Method       | LPIPS↓   | SSIM↑    | PSNR↑    |
> |--------------|---------|---------|---------|
> | w/o DCTA     | 0.1902  | 0.8779  | 25.68   |
> | **w/ DCTA**  | **0.1734** | **0.8885** | **26.45** |
>
> ### **Q5: Memory Usage Analysis.**
> **A5:** Thank you for your constructive suggestion. Here, we report the VRAM consumption of the vanilla DiT model and DiCache when using cached residuals of different orders. Experiments in this section are conducted with the T2V model HunyuanVideo ($544\times960$ resolution, $129$ frames, $50$ sampling steps) on an NVIDIA A800 80GB GPU. To achieve faster inference, CPU offloading is disabled. As shown in the table below, although using higher-order cached residuals in DiCache leads to a slight increase in VRAM usage, the overall additional consumption remains modest because the cached residual tensor is relatively small.
>
> **Table4: VRAM Consumption of HunyuanVideo under Different Cache Orders**
>
> | Cache Order | VRAM (GB) |
> |:---------------|:--------------:|
> | Vanilla HunyuanVideo | 60.51 |
> | 0 (DiCache w/o DCTA) | 61.66 |
> | 1 (DiCache w/ DCTA)  | 62.82 |
> | 2           | 63.59     |
> | 4           | 65.13     |
>
> ### **Q6: Reasoning on probing.**
> **A6:** Thank you for your valuable suggestion. For flow models (e.g., Flux, HunyuanVideo, WAN series), the output feature of their transformers $f_t^M$ is decoded using an MLP $\texttt{MLP}_\text{out}$ to obtain the model’s prediction of the flow velocity at timestep $t$, which is the result of processing and refining the noisy input $x_t$ in the latent space through $M$ DiT blocks. This process can be formulated as:
>
> $$v_\theta^{t,M} = \texttt{MLP}_\text{out}(f_t^M),$$
>
> in which $v_\theta^{t,M}$ denotes the velocity prediced by the entire DiT with $M$ blocks.
> Inspired by developments in LLMs [1], the output feature of the $m$-th ($m\in[1,M]$) DiT block $f_t^m$ can also be decoded to a intermediate velocity $v_\theta^{t,m}$ with $\texttt{MLP}_\text{out}$:
>
> $$ v_\theta^{t,m} = \texttt{MLP}_\text{out}(f_t^m),$$
>
> where $v_\theta^{t,m}$ can be interpreted as the velocity predicted by the first $m$ DiT blocks. Since $v_\theta^{t,M}$ is obtained by further refining $v_\theta^{t,m}$ through the remaining $M - m$ DiT blocks, **a well-trained DiT is expected to produce smooth transitions of flow velocity across blocks. In other words, $v_\theta^{t,M}$ and $v_\theta^{t,m}$ should exhibit consistent and similar evolution trends**.
>
> To obtain the semantic information represented by $v_\theta^{t,m}$, we compute its corresponding estimated clean sample $\hat{x}_0^{t,m}$:
>
> $$\hat{x} _ 0^{t,m} = x_t - v_\theta^{t,M}t,$$
>
> which can be further decoded to the pixel space using VAE:
>
> $$I_t^m = \texttt{VAE}(\hat{x}_0^{t,m}),$$
>
> where $I_t^m$ represent the estimated clean image decoded from $v_\theta^{t,m}$. A visualization of $I_t^m$ with different $m$ on Flux can be found in Figure 21 in the revised manuscript. As can be observed from these images, $I_t^m$ evolve smoothly as $m$ increases, where early blocks capture coarse semantic structures and later blocks progressively refine details. This phenomenon provides strong evidence that the intermediate velocities $v_\theta^{t,m}$ exhibit consistent evolution trends across blocks, ultimately converging toward the final prediction $v_\theta^{t,M}$. Such smooth and semantically aligned transitions provide clear evidence that shallow-layer DiT features already contain reliable semantic cues, thereby justifying the effectiveness of shallow-layer probing in our DiCache framework.
>
> ### **Q7: Minor typos in the manuscript.**
> **A7:** Thank you for your valuable comment, and we sincerely apologize for any misunderstanding that may have been caused. We have fixed these typos in the revised manuscript.
>
> ### **Reference**
>
> [1] Chuang Y S, Xie Y, Luo H, et al. Dola: Decoding by contrasting layers improves factuality in large language models[J]. arXiv preprint arXiv:2309.03883, 2023.

---

> ### Comment · Reviewer_HkJV · 2025-11-24
>
> I appreciate the authors’ additional experiments and clarifications addressing the weaknesses and questions I previously raised.
>
> In particular, the responses to Weaknesses 1 and 5, as well as the questions, were clear and informative.
>
> However, I still find that the overall contributions remain modest, and the effectiveness of DCTA appears marginal. In addition, the ablation studies on the score’s mean and variance are not sufficient to demonstrate that DiCache consistently outperforms alternative methods under varied conditions.
>
> While I believe this paper is well conducted work, I will maintain my original score, primarily due to the former reasons.

---

> > ### Author Response · Authors · 2025-11-24
> >
> > Dear Reviewer HkJV,
> >
> > Thank you for your constructive feedback on our work!
> >
> > We are glad to hear that our rebuttal has addressed the weaknesses and questions you raised, and we appreciate your continued recognition of our work for its novelty in dynamically aligning cache trajectories and its clear presentation.
> >
> > We would also like to claim that, through the carefully designed Online Probe Profiling Scheme and Dynamic Cache Trajectory Alignment, our DiCache surpasses existing state-of-the-art methods in both qualitative and quantitative evaluations, and provides a viable paradigm for future research.
> >
> > We sincerely appreciate your time and effort in providing such meticulous reviews and insightful comments.
> >
> > Best regards,
> >
> > Authors

---

### Official Review · Reviewer_bWWH · 2025-11-01

**Soundness:** 3
**Presentation:** 3
**Contribution:** 3
**Rating:** 6
**Confidence:** 4

**Summary:**

This paper introduces DiCache, a training-free, adaptive caching strategy to accelerate diffusion models. DiCache addresses two core challenges in caching-based acceleration: "when to cache" and "how to use cache." The method introduces two components: (1) an Online Probe Profiling Scheme to dynamically determine when to reuse cached outputs, and (2) Dynamic Cache Trajectory Alignment to improve multi-step feature reuse through shallow-to-deep trajectory consistency. Extensive experiments on WAN 2.1, HunyuanVideo, and Flux validate that DiCache achieves significant speedup while maintaining high visual fidelity.

**Strengths:**

* The motivation behind the proposed method is well articulated. The method's design is strongly supported by empirical evidence presented in the paper.

* The method is completely training-free, making it highly practical and broadly applicable across different diffusion models.

**Weaknesses:**

* The proposed reuse threshold δ appears to require manual, per-model tuning (δ = 0.2 for WAN 2.1, δ = 0.1 for HunyuanVideo, δ = 0.4 for Flux), which may reduce generality and increase tuning effort for new architectures.

* While the probe is “shallow,” it is still computed at every single timestep to accumulate caching error. It remains unclear how much speedup is offset by repeated probing on large backbones.

**Questions:**

1.Given the memory and dynamic variability challenges, is there a possibility to automatically determine δ or adaptively calibrate it without per-model tuning?

2.The probe is shallow but still executed at every timestep. Could the authors provide a detailed comparison of the probe cost across different architectures.

3.The probe uses L1 relative distance on shallow features. Have alternative feature distance metrics been considered, especially for semantic coherence?

---

> ### Author Response · Authors · 2025-11-19
> **Response to Reviewer bWWH (1/2)**
>
> We are greatly encouraged that you found our work well-motivated, highly practical, and broadly applicable across different diffusion models. We sincerely thank you for your valuable suggestions, which certainly help improve our work. We have accordingly refined our paper as follows:
>
> ### **Q1: Given the memory and dynamic variability challenges, is there a possibility to automatically determine δ or adaptively calibrate it without per-model tuning?**
>
> **A1:** Thank you for your constructive suggestion.
> Utilizing a threshold for cache reusing is a broadly applicable approach adopted by previous outstanding works, including TeaCache, AdaCache, and EasyCache. Admittedly, the recommended value of the reuse threshold $\delta$ varies across models, but the differences are relatively minor. These variations arise from the differences among DiT models with different training data and architectures (e.g., Flux is a T2I model with 57 layers and 12B parameters, while WAN 2.1-1.3B is a T2V model with 30 layers and 1.3B parameters). A potential way to calibrate $\delta$ for a new model is to perform a simple one-dimensional sweep over a commonly used threshold range (e.g., a $\delta \in [0.1, 0.5]$ is recommended since it has shown effectiveness across most tested models) using a step size of 0.05 or 0.1. **Furthermore, once the value of $\delta$ is determined, it exhibits strong stability across different samples within the same model.** Specifically, we sample 1K diverse prompts (the same evaluation set as the main paper) from the LAION-5B dataset and use Flux for image generation. The results demonstrate that DiCache not only maintains the lowest LPIPS but also exhibits the lowest LPIPS variance compared to state-of-the-art methods (as shown in the table below), indicating its stable performance across diverse samples and strong generalization capability.
>
> **Table1: Mean and Variance of LPIPS on Flux**
>
> | Method       | Mean(LPIPS)↓ | Variance(LPIPS)↓ |
> |:---------------|:--------------:|:------------------:|
> | TeaCache     | 0.4053       | 0.010            |
> | TaylorSeer   | 0.4709       | 0.007            |
> | EasyCache    | 0.3049       | 0.008            |
> | **DiCache (Ours)** | **0.2704** | **0.004**       |
>
> In addition, since **acceleration for diffusion models is a topic highly dependent on practical needs**, some users may prioritize higher speedups, while others may prefer better visual fidelity. Therefore, adjusting $\delta$ at inference time according to practical needs provides greater flexibility and better accommodates diverse user requirements.
>
> ### **Q2: The probe is shallow but still executed at every timestep. Could the authors provide a detailed comparison of the probe cost across different architectures.**
> **A2:** Thanks for your valuable suggestion. As we analyzed in Section 3.2 of the main paper, DiCache supports resuming computation from the probe layer. Therefore, **the probing cost is only incurred for the skipped steps**. Here, we provide a detailed analysis and comparison of the probing cost. Assuming the model has a total of $N$ sampling steps, the skipping rate after applying DiCache is denoted as $\alpha$. The number of steps that the model needs to compute ($N_\text{compute}$) and the number of skipped steps ($N_\text{skip}$) are given by:
>
> $$N_\text{compute} = (1-\alpha)N, $$
>
> $$N_\text{skip} = \alpha N.$$
>
> Since the probing cost is only incurred for the skipped steps, following the definitions in the main paper, let $M$ denote the total number of layers in the DiT model, $m$ the probe depth, and further let $t$ represent the inference time per step, the total probing cost during the sampling process can be expressed as:
>
> $$T_\text{probe} = N_\text{skip} \times \frac{m}{M} \times t = \frac{\alpha Nmt}{M}.$$
>
> Following our main experiments, the values of the quintuple $(N, \alpha, m, M, t)$ for different models are: $(30, 0.70, 1, 57, 0.50s)$ for Flux, $(50, 0.62, 1, 30, 3.85s)$ for WAN 2.1, $(50, 0.58, 1, 60, 23.73s)$ for HunyuanVideo. These values were obtained on an NVIDIA A800 80GB GPU. By substituting them into the above equation, we can compute the probing cost $T_\text{probe}$ for different models, as shown in the table below. Moreover, we also calculate the ratio of the probing cost to the total inference time with DiCache ($T_\text{probe}/T_\text{all}$). In summary, since DiT models are typically deep while DiCache introduces only shallow-layer probing, the overall additional cost incurred by the probing operation remains minimal.
>
> **Table2: Probing Cost of Different Models**
>
> | Model          | $T_\text{probe}$ (sec) | $T_\text{probe}/T_\text{all}$ |
> |:---------------|:--------------:|:------------------:|
> | Flux           | 0.19                     | 4%                              |
> | WAN 2.1-1.3B   | 3.98                     | 5%                              |
> | HunyuanVideo   | 11.47                    | 2%                              |

---

> > ### Comment · Reviewer_bWWH · 2025-11-20
> >
> > For Q1:  The response looks good. I strongly suggest adding the results in the revision.
> >
> > For Q2: The Probing Cost occupies very low ratios across different video models. That shows the efficiency of the proposed method.
> >
> > Based on these, I think the authors make a good rebuttal, so i recommend to accept this work.

---

> > > ### Author Response · Authors · 2025-11-20
> > >
> > > Dear Reviewer bWWH,
> > >
> > > Thank you for your positive and encouraging feedback on our work!
> > >
> > > We are glad to hear that our rebuttal has addressed your concerns. The corresponding revisions will be incorporated into the final version of the paper.
> > >
> > > We sincerely appreciate your time and effort in providing such meticulous reviews and insightful comments.
> > >
> > > Best regards,
> > >
> > > Authors

---

> ### Author Response · Authors · 2025-11-19
> **Response to Reviewer bWWH (2/2)**
>
> ### **Q3: The probe uses L1 relative distance on shallow features. Have alternative feature distance metrics been considered, especially for semantic coherence?**
> **A3:** Thank you for your insightful comment. Here, we derive a semantic distance that could potentially be used as an alternative caching error metric. For flow models (e.g., Flux, HunyuanVideo, WAN series), the output feature of their transformers $f_t^M$ is decoded using an MLP $\texttt{MLP}_\text{out}$ to obtain the model’s prediction of the flow velocity at timestep $t$, which is the result of processing and refining the noisy input $x_t$ in the latent space through $M$ DiT blocks. This process can be formulated as:
>
> $$v_\theta^{t,M} = \texttt{MLP}_\text{out}(f_t^M),$$
>
> in which $v_\theta^{t,M}$ denotes the velocity prediced by the entire DiT with $M$ blocks.
> Inspired by developments in LLMs [1], the output feature of the $m$-th ($m\in[1,M]$) DiT block $f_t^m$ can also be decoded to an intermediate velocity $v_\theta^{t,m}$ with $\texttt{MLP}_\text{out}$:
>
> $$v_\theta^{t,m} = \texttt{MLP}_\text{out}(f_t^m),$$
>
> where $v_\theta^{t,m}$ can be interpreted as the velocity predicted by the first $m$ DiT blocks. To obtain the semantic information represented by $v_\theta^{t,m}$, we compute its corresponding estimated clean sample $\hat{x}_0^{t,m}$:
>
> $$\hat{x} _ 0^{t,m} = x_t - v_\theta^{t,M}t,$$
>
> which can be further decoded to the pixel space using VAE:
>
> $$I_t^m = \texttt{VAE}(\hat{x}_0^{t,m}),$$
>
> where $I_t^m$ represents the estimated clean image decoded from $\hat{x}_0^{t,m}$. A visualization of $I_t^m$ on Flux can be found in Figure 20 in the revised manuscript. Since $I_t^m$ lies in the pixel space and contains semantic information, perceptual metrics such as LPIPS can be employed to measure the semantic distance between these images at consecutive timesteps:
>
> $$\text{Dist} _ \text{semantic}^{t} = \text{LPIPS}(I_t^m, I_{t+1}^m),$$
>
> in which $\text{Dist}_\text{semantic}^{t}$ can serve as a potential caching error indicator for DiCache.
> However, decoding with VAE is quite time-consuming, especially for video models (for example, on an NVIDIA A100 80GB GPU, the VAE decode time for HunyuanVideo with $544\times960$ resolution and $129$ frames is 30.86$s$, which is even longer than performing a single inference step of HunyuanVideo). Therefore, this semantic distance is not adopted as the caching error metric in our framework. In contrast, the adopted L1 relative distance serves as a lightweight choice that bypasses additional computational overhead.
>
> ### **Reference**
>
> [1] Chuang Y S, Xie Y, Luo H, et al. Dola: Decoding by contrasting layers improves factuality in large language models[J]. arXiv preprint arXiv:2309.03883, 2023.

---

### Author Response · Authors · 2025-11-19
**Official Comment by Authors**

We sincerely appreciate all reviewers for their time and efforts in the review. We are highly encouraged that: Reviewer bWWH found our work **well-motivated, highly practical**, and  **broadly applicable across different models**; Reviewer HkJV described our work as **novel and effective** in dynamically aligning cache trajectories, **outperforming prior caching methods** without sacrificing quality, and clear in presentation; Reviewer MEuZ found our work **addressing the two fundamental challenges of cache-based acceleration**, compatible with other acceleration techniques, and well-grounded in feature observations; Reviewer 1sUf described our work as clearly motivated, **having convincing empirical findings**, especially useful for video generation, and **easy to integrate with other intra-step acceleration methods** like Sparse Attention. These comments have greatly motivated us, which certainly help improve our work. Based on their valuable and insightful suggestions, we have carefully revised our manuscript, as detailed below:

1. We have **included additional quantitative metrics to provide a more comprehensive evaluation**. Specifically, we added VBench [1] for T2V evaluation and incorporated CLIP Score [2] and ImageReward [3] for T2I evaluation. Moreover, we conducted a user study to further assess human-perceived quality.

2. We have **provided a detailed explanation and analysis regarding the two hyperparameters** of DiCache.

3. We have **clarified why shallow-layer probing is effective in DiT models** and provided detailed derivations, visualizations and additional diagrams to support it.

4. We have **performed additional analysis and ablation studies on Dynamic Cache Trajectory Alignment (DCTA)** to further demonstrate its lightweight design and effectiveness across different models.

5. We have **added the comparison with more methods**, including FasterCache [4], FORA [5], and $\Delta$-DiT [6].

6. We have **provided a detailed comparison of the probing cost across different architectures**.

7. We have **discussed a potentially viable semantic distance metric** as an alternative to the L1 relative feature distance currently used in DiCache.

8. We have **validated the effectiveness of the proposed DiCache on the distilled DiT model**.

9. We have **clarified previously ambiguous descriptions**, including redesigning the pipeline, adding missing abbreviations, enlarging the axes in several figures, and highlighting the novelty of our approach compared with existing methods.

10. We have **fixed minor typos** in the manuscript.

The detailed point-by-point responses are listed in the columns below.

**Reference**

[1] Huang Z, He Y, Yu J, et al. Vbench: Comprehensive benchmark suite for video generative models[C]//Proceedings of the IEEE/CVF Conference on Computer Vision and Pattern Recognition. 2024: 21807-21818.

[2] Radford A, Kim J W, Hallacy C, et al. Learning transferable visual models from natural language supervision[C]//International conference on machine learning. PmLR, 2021: 8748-8763.

[3] Xu J, Liu X, Wu Y, et al. Imagereward: Learning and evaluating human preferences for text-to-image generation[J]. Advances in Neural Information Processing Systems, 2023, 36: 15903-15935.

[4] Lv Z, Si C, Song J, et al. Fastercache: Training-free video diffusion model acceleration with high quality[J]. arXiv preprint arXiv:2410.19355, 2024.

[5] Selvaraju P, Ding T, Chen T, et al. Fora: Fast-forward caching in diffusion transformer acceleration[J]. arXiv preprint arXiv:2407.01425, 2024.

[6] Chen P, Shen M, Ye P, et al. $\Delta $-DiT: A Training-Free Acceleration Method Tailored for Diffusion Transformers[J]. arXiv preprint arXiv:2406.01125, 2024.

---

### Author Response · Authors · 2025-11-29

We sincerely appreciate ACs, SACs, PCs and all reviewers for their time and efforts in the review. **During the first two weeks of the discussion period, we have received positive ratings from all four reviewers,** which have been very motivating for us.

Specifically, **our scores increased from 6, 6, 6, 4 to 6, 6, 6, 6**. Reviewer bWWH noted that "I think the authors make a good rebuttal, so i recommend to accept this work."; Reviewer HkJV found our response "clear and informative", and described our paper as "a well-conducted work"; Reviewer 1sUf stated that "Most of my concerns have been addressed" and raised the score accordingly.

Based on their constructive feedbacks, we believe that most concerns of the reviewers have been addressed by our rebuttal. The revisions will be incorporated into the final version. Thanks again to ACs, SACs, PCs and all the reviewers for their consistent and dedicated contributions throughout the review process.

---

### Meta-Review · Area_Chair_JErZ · 2026-01-06

**Summary:**

This paper introduces DiCache, a training-free adaptive caching framework that accelerates diffusion model inference by dynamically deciding when and how to reuse cached features. The method uses a shallow-layer online probe to estimate caching errors in real time and aligns deep-layer outputs via trajectory consistency. Experiments on models like WAN 2.1 and Flux demonstrate speedups of 2.34×–3.22× while maintaining or improving visual quality compared to strong baselines.

Reviewers initially raised concerns about hyperparameter sensitivity, novelty relative to prior caching works, the marginal gain from DCTA, and evaluation completeness. In the rebuttal, the authors provided extensive additional experiments—including VBench scores, user studies, memory profiling, and comparisons with newer methods—and clarified the theoretical foundation of shallow deep feature correlation in DiTs. These responses satisfied the reviewers, leading all four scores to settle at 6 (marginally above acceptance).

Overall, the paper presents a well‑motivated, practical, and plug‑and‑play solution for diffusion‑model acceleration.

**Reviewer Concerns:**

Initially, the submission received mixed but overall positive scores (6, 6, 6, 4). Reviewers raised concerns mainly regarding evaluation completeness, hyperparameter sensitivity, probing overhead, and baseline coverage. These issues were carefully addressed in the rebuttal with additional experiments and clarifications.Presentation (Addressed): Reviewer 1sUf suggested redesigning the pipeline diagrams. The authors provided a clearer, redesigned version of the framework overview in the appendix.

To address concerns about metric comprehensiveness, the authors added broader perceptual evaluations, including VBench for video models, CLIP Score and ImageReward for image models, as well as a user study with 30 participants, all of which consistently demonstrate DiCache’s superior performance. In response to baseline-related concerns, comparisons with FasterCache, FORA, and Δ-DiT were added, showing clear advantages in both efficiency and generation quality.

Concerns regarding the manual tuning of the reuse threshold δ were alleviated by showing that δ remains stable across samples within the same model, with sensitivity analyses indicating a shared effective range (δ ∈ [0.1, 0.5]) across different architectures. Additionally, the authors analyzed the probing overhead in detail, demonstrating that probing accounts for only 2%–5% of total inference time, confirming the practicality of the proposed approach.

**Reviewer Scores:**

The final consensus among reviewers became solidly positive, with scores converging to 6, 6, 6, 6. Reviewer 1sUf raised their score from a 4 to a 6 after the rebuttal. Reviewer bWWH recommended acceptance, noting the rebuttal was strong and demonstrated efficiency. Reviewer HkJV maintained a 6, acknowledging the work was well-conducted while personally viewing the impact of DCTA as marginal.

---

### Decision · Program_Chairs · 2026-01-26

Accept (Poster)